

# A comprehensive molecular phylogeny of Geometridae (Lepidoptera) with a focus on enigmatic small subfamilies

Leidys Murillo-Ramos[1,2], Gunnar Brehm[3], Pasi Sihvonen[4],
Axel Hausmann[5], Sille Holm[6], Hamid Reza Ghanavi[2], Erki Õunap[6,7],
Andro Truuverk[8], Hermann Staude[9], Egbert Friedrich[10],
Toomas Tammaru[6] and Niklas Wahlberg[2]

[1] Grupo Biología Evolutiva, Department of Biology, Universidad de Sucre, Sincelejo, Sucre, Colombia
[2] Systematic Biology Group, Department of Biology, Lund University, Lund, Sweden
[3] Institut für Zoologie und Evolutionsbiologie, Phyletisches Museum, Jena, Germany
[4] Finnish Museum of Natural History, University of Helsinki, Helsinki, Finland
[5] Staatliche Naturwissenschaftliche Sammlungen Bayerns, München, Germany
[6] Department of Zoology, Institute of Ecology and Earth Sciences, University of Tartu, Tartu, Estonia
[7] Estonian University of Life Sciences, Institute of Agricultural and Environmental Sciences, Tartu, Estonia
[8] Natural History Museum, University of Tartu, Tartu, Estonia
[9] LepsocAfrica, Magaliesburg, South Africa
[10] Berghoffsweg 5, Jena, Germany

Corresponding authors
Leidys Murillo-Ramos,
leidys.murillo@unisucre.edu.co
Niklas Wahlberg,
niklas.wahlberg@biol.lu.se

## ABSTRACT

Our study aims to investigate the relationships of the major lineages within the moth family Geometridae, with a focus on the poorly studied Oenochrominae-Desmobathrinae complex, and to translate some of the results into a coherent subfamilial and tribal level classification for the family. We analyzed a molecular dataset of 1,206 Geometroidea terminal taxa from all biogeographical regions comprising up to 11 molecular markers that includes one mitochondrial (COI) and 10 protein-coding nuclear gene regions (wingless, ArgK, MDH, RpS5, GAPDH, IDH, Ca-ATPase, Nex9, EF-1alpha, CAD). The molecular data set was analyzed using maximum likelihood as implemented in IQ-TREE and RAxML. We found high support for the subfamilies Larentiinae, Geometrinae and Ennominae in their traditional scopes. Sterrhinae becomes monophyletic only if *Ergavia* Walker, *Ametris* Hübner and *Macrotes* Westwood, which are currently placed in Oenochrominae, are formally transferred to Sterrhinae. Desmobathrinae and Oenochrominae are found to be polyphyletic. The concepts of Oenochrominae and Desmobathrinae required major revision and, after appropriate rearrangements, these groups also form monophyletic subfamily-level entities. Oenochrominae *s.str.* as originally conceived by Guenée is phylogenetically distant from *Epidesmia* and its close relatives. The latter is hereby described as the subfamily Epidesmiinae Murillo-Ramos, Sihvonen & Brehm, **subfam. nov.** Epidesmiinae are a lineage of "slender-bodied Oenochrominae" that include the genera *Ecphyas* Turner, *Systatica* Turner, *Adeixis* Warren, *Dichromodes* Guenée, *Phrixocomes* Turner, *Abraxaphantes* Warren, *Epidesmia* Duncan & Westwood and *Phrataria* Walker. Archiearinae are monophyletic when *Dirce* and *Acalyphes* are formally transferred to Ennominae. We
also found that many tribes were para- or polyphyletic and therefore propose tens of taxonomic changes at the tribe and subfamily levels. Archaeobalbini **stat. rev.** Viidalepp (Geometrinae) is raised from synonymy with Pseudoterpnini Warren to tribal rank. Chlorodontoperini Murillo-Ramos, Sihvonen & Brehm, **trib. nov.** and Drepanogynini Murillo-Ramos, Sihvonen & Brehm, **trib. nov.** are described as new tribes in Geometrinae and Ennominae, respectively.

## INTRODUCTION

Geometridae are the second most species-rich family of Lepidoptera, with approximately 24,000 described species (number from *Van Nieukerken et al. (2011)* updated by the authors) found in all regions except Antarctica. The monophyly of Geometridae is well supported based on distinctive morphological characters (*Cook & Scoble, 1992*; *Scoble, 1992*; *Minet & Scoble, 1999*). In particular, adult members of the family possess paired tympanal organs at the base of the abdomen, while in larvae the prolegs are reduced to two pairs in almost all species, which causes the larvae to move in a looping manner (*Minet & Scoble, 1999*).

The phylogenetic relationships of the major subdivisions of Geometridae have been studied based on molecular data, which have contributed to the understanding of the evolutionary relationships within the family (*Abraham et al., 2001*; *Yamamoto & Sota, 2007*; *Sihvonen et al., 2011*). Eight subfamilies are currently recognized in Geometridae (*Sihvonen et al., 2011*). Several recent molecular and morphological studies have attempted to confirm the monophyly or clarify the taxonomy of most of these groups, for instance: Sterrhinae (*Holloway, 1997*; *Hausmann, 2004*; *Sihvonen & Kaila, 2004*; *Õunap, Viidalepp & Saarma, 2008*), Larentiinae (*Holloway, 1997*; *Mironov, 2003*; *Viidalepp, 2006, 2011*; *Hausmann & Viidalepp, 2012*; *Õunap, Viidalepp & Truuverk, 2016*), Desmobathrinae (*Holloway, 1996*; *Hausmann, 2001*), Archiearinae (*Hausmann, 2001*; *Young, 2006*), Oenochrominae (*Holloway, 1996*; *Scoble & Edwards, 1990*; *Cook & Scoble, 1992*; *Hausmann, 2001*; *Young, 2006*), Geometrinae (*Cook et al., 1994*; *Pitkin, 1996*; *Hausmann, 2001*; *Ban et al., 2018*), Orthostixinae (*Holloway, 1997*) and Ennominae (*Holloway, 1994*; *Pitkin, 2002*; *Beljaev, 2006*; *Young, 2006*; *Wahlberg et al., 2010*; *Õunap et al., 2011*; *Skou & Sihvonen, 2015*; *Sihvonen, Staude & Mutanen, 2015*), but questions remain. An important shortcoming is that our understanding of geometrid systematics is biased towards the long-studied European fauna, whereas the highest diversity of this family is in the tropics, which are still largely unexplored (*Brehm et al., 2016*). Many species remain undescribed and there are many uncertainties in the classification of tropical taxa.

One of the most comprehensive phylogenetic studies on Geometridae to date was published by *Sihvonen et al. (2011)*. They analyzed a data set of 164 taxa and up to eight genetic markers, and the most species-rich subfamilies were confirmed as monophyletic. However, the systematic positions of Oenochrominae and Desmobathrinae remained uncertain due to low taxon sampling and genetic markers, and both subfamilies were

suspected to be polyphyletic. Moreover, because of taxonomic uncertainty, many geometrid genera, especially among tropical taxa, remained unassigned to any tribe.

This study is the first in a series of papers that investigate the phylogenetic relationships of Geometridae on the basis of global sampling. Our dataset comprises 1,192 terminal taxa of Geometridae and 14 outgroup taxa, with samples from all major biomes, using up to 11 molecular markers. Our paper includes an overview of the relationships of the major lineages within the family, with the particular aim of defining the limits and finding the phylogenetic affinities of the subfamilies, with a focus on Oenochrominae and Desmobathrinae. Further papers in the series will focus on particular subfamilies and regions, and will build upon the taxonomic changes proposed in the present article: e.g., relationships in Sterrhinae (P. Sihvonen et al., 2019, unpublished data), New World taxa (G. Brehm et al., 2019, unpublished data), Larentiinae (E. Õunap et al., 2019, unpublished data) and the ennomine tribe Boarmiini (L. Murillo-Ramos et al., 2019, unpublished data).

Oenochrominae and Desmobathrinae are considered the most controversial subfamilies in Geometridae. A close relationship of these subfamilies has been proposed both in morphological (*Meyrick, 1889*; *Cook & Scoble, 1992*; *Holloway, 1996*) and in molecular studies (*Sihvonen et al., 2011*; *Ban et al., 2018*). In early classifications, species of Desmobathrinae and Oenochrominae were classified in the family Monocteniadae (*Meyrick, 1889*), which is currently considered a junior synonym of Oenochrominae Guenée. Meyrick diagnosed them on the basis of the position of the R veins in the hindwing and Sc+R1 in the forewing (*Scoble & Edwards, 1990*). However, the classification proposed by Meyrick was not fully supported by subsequent taxonomic work (*Scoble & Edwards, 1990*; *Cook & Scoble, 1992*; *Holloway, 1996*). Too often, Oenochrominae was used for geometrids that could not be placed in other subfamilies, and at some point, even included Hedylidae, the moth-butterflies (*Scoble, 1992*). Unsurprisingly, many taxa formerly classified in Oenochrominae have recently been shown to be misplaced (*Holloway, 1997*; *Staude, 2001*; *Sihvonen & Staude, 2011*; *Staude & Sihvonen, 2014*). In *Scoble & Edwards (1990)*, the family concept of Oenochrominae was restricted to the robust-bodied Australian genera, with one representative from the Oriental region. *Scoble & Edwards (1990)* were not able to find synapomorphies to define Monocteniadae *sensu* Meyrick, and referred back to the original grouping proposed by *Guenée (1858)*. They restricted Oenochrominae to a core clade based on male genitalia: the diaphragm dorsal to the anellus is fused with the transtilla to form a rigid plate. Additionally, *Cook & Scoble (1992)* suggested that the circular form of the lacinia and its orientation parallel to the tympanum was apomorphic for these robust-bodied Oenochrominae.

In an extensive morphological study, *Holloway (1996)* delimited the subfamily Desmobathrinae to include species with slender appendages and bodies previously assigned to Oenochrominae. According to *Holloway (1996)*, Desmobathrinae comprises two tribes: Eumeleini and Desmobathrini. However, no synapomorphies were found to link the two tribes. *Holloway (1996)* noted that the modification of the tegumen of the male genitalia was variable in both groups but that the reduction of cremastral spines in the pupa from eight to four in *Ozola* Walker, 1861 and *Eumelea* Duncan & Westwood, 1841 provided evidence of a close relationship between Eumeleini and Desmobathrini.

Currently, 328 species (76 genera) are included in Oenochrominae, and 248 species (19 genera) are assigned to Desmobathrinae (*Beccaloni et al., 2003*; *Sihvonen et al., 2011*; *Sihvonen, Staude & Mutanen, 2015*).

Most recent molecular phylogenies have shown Oenochrominae and Desmobathrinae to be intermingled (*Sihvonen et al., 2011*; *Ban et al., 2018*), but previous taxon sampling was limited to eight and four species, respectively. The poor taxon sampling and unresolved relationships around the oenochromine and desmobathrine complex called for additional phylogenetic studies to clarify the relationships of these poorly known taxa within Geometridae. We hypothesize that both Oenochrominae and Desmobathrinae are para- or polyphyletic assemblages, and we address this hypothesis with studying 29 terminal taxa of Oenochrominae and 11 representatives of Desmobathrinae, mostly from the Australian and Oriental Regions.

## MATERIALS AND METHODS

### Material acquisition, taxon sampling and species identification

In addition to 461 terminal taxa with published sequences (see Data S1), we included sequences from 745 terminal taxa in our study (Data S1). Representative taxa of all subfamilies recognized in Geometridae were included, except for the small subfamily Orthostixinae for which most molecular markers could not be amplified successfully. A total of 93 tribes are represented in this study following recent phylogenetic hypotheses and classifications (*Sihvonen et al., 2011*; *Wahlberg et al., 2010*; *Sihvonen, Staude & Mutanen, 2015*; *Õunap, Viidalepp & Truuverk, 2016*; *Ban et al., 2018*). In addition, 14 non-geometrid species belonging to other families of Geometroidea were included as outgroups based on the hypothesis proposed by *Regier et al. (2009*, *2013)*. Where possible, two or more samples were included per tribe and genus, especially for species-rich groups that are widely distributed and in cases where genera were suspected to be poly- or paraphyletic. We emphasized type species or species similar to type species, judged by morphological characters and/or genetic similarity of DNA barcodes in order to better inform subsequent taxonomic work, to favor nomenclatorial stability and to establish the phylogenetic positions of genera unassigned to tribes.

Sampled individuals were identified by the authors using appropriate literature, by comparing them with type material from different collections, museums and DNA barcode sequences. Moreover, we compiled an illustrated catalog of all Archiearinae, Desmobathrinae and Oenochrominae taxa included in this study, to demonstrate their morphological diversity and to facilitate subsequent verification of our identifications. This catalog contains images of all analyzed specimens of the above-mentioned taxa as well as photographs of the respective type material (Data S2). Further taxa from other subfamilies will be illustrated in other papers (G. Brehm et al., 2019, unpublished data, P. Sihvonen et al., 2019, unpublished data, E. Õunap et al., 2019, unpublished data). Some of the studied specimens could not yet be assigned to species, and their identifications are preliminary, particularly for (potentially undescribed) tropical species. Taxonomic data, voucher IDs, number of genes, current systematic placement and references to relevant literature with regard to tribal assignment, are shown in Data S1.

## Molecular techniques

DNA was extracted from one to three legs of specimens either preserved in ethanol or dry. In a few cases, other sources of tissue were used, such as parts of larvae. The remaining parts of specimens were preserved as vouchers deposited in the collections of origin, both public and private (eventually private material will be deposited in public museum collections). Genomic DNA was extracted and purified using a NucleoSpin® Tissue Kit (MACHEREY-NAGEL, Düren, Germany), following the manufacturer's protocol. DNA amplification and sequencing were carried out following protocols proposed by *Wahlberg & Wheat (2008)* and *Wahlberg et al. (2016)*. PCR products were visualized on agarose gels. PCR products were cleaned enzymatically with Exonuclease I and FastAP Thermosensitive Alkaline Phosphatase (ThermoFisher Scientific, Waltham, MA, USA) and sent to Macrogen Europe (Amsterdam, Netherlands) for Sanger sequencing. One mitochondrial (*cytochrome oxidase subunit I*, COI) and 10 protein-coding nuclear gene regions, *carbamoylphosphate synthetase* (CAD), *Ribosomal Protein S5* (RpS5), *wingless* (wgl), *cytosolic malate dehydrogenase* (MDH), *glyceraldehydes-3-phosphate dehydrogenase* (GAPDH), *Elongation factor 1 alpha* (EF-1alpha), *Arginine Kinase* (ArgK), *Isocitrate dehydrogenase* (IDH), *sorting nexin-9-like* (Nex9) and *sarco/endoplasmic reticulum calcium ATPase* (Ca-ATPase), were sequenced. To check for potential misidentifications, DNA barcode sequences were compared to those in BOLD (*Ratnasingham & Hebert, 2007*) where references of more than 21,000 geometrid species are available, some 10,000 of them being reliably identified to Linnean species names (*Ratnasingham & Hebert, 2007*). GenBank accession numbers for sequences used in this study are provided in Data S1.

## Alignment and cleaning sequences

Multiple sequence alignments were carried out in MAFFT as implemented in Geneious v.11.0.2 (Biomatters, http://www.geneious.com/) for each gene based on a reference sequence of Geometridae downloaded from the database VoSeq (*Peña & Malm, 2012*). The alignment of each gene was carefully checked by eye relative to the reference sequence, taking into account the respective genetic codes and reading frames. Heterozygous positions were coded with IUPAC codes. Sequences with bad quality were removed from the alignments. Aligned sequences were uploaded to VoSeq (*Peña & Malm, 2012*) and then assembled into a dataset comprising 1,206 taxa. The final dataset had a concatenated length of 7665 bp including gaps. To check for possible errors in alignments, potentially contaminated or identical sequences and misidentifications, we constructed maximum-likelihood trees for each gene. These preliminary analyses were conducted using RAxML-HPC2 V.8.2.10 (*Stamatakis, 2014*) on the web-server CIPRES Science Gateway (*Miller, Pfeiffer & Schwartz, 2010*). The final data set included at least three genes per taxon except for *Oenochroma vinaria* (Guenée, 1858), *Acalyphes philorites* Turner, 1925, *Dirce lunaris* (Meyrick, 1890), *D. aesiodora* Turner, 1922, *Furcatrox australis* (Rosenstock, 1885), *Chlorodontopera mandarinata* (Leech, 1889), *Chlorozancla falcatus* (Hampson, 1895), *Pamphlebia rubrolimbraria* (Guenée, 1858) and *Thetidia albocostaria* (Bremer, 1864). For these taxa, included in studies by *Young (2006)* and *Ban et al. (2018)*, only two markers were available. The final data matrix included 32% missing data.

## Tree search strategies and model selection

We ran maximum likelihood analyses with a data set partitioned by gene and codon position using IQ-TREE V1.6.10 (*Nguyen et al., 2015*) and data partitioned by codon in RAxML (*Stamatakis, 2014*). Best-fitting substitution models were selected by ModelFinder, which is a model-selection method that incorporates a model of flexible rate heterogeneity across sites (*Kalyaanamoorthy et al., 2017*). ModelFinder implements a greedy strategy as implemented in PartitionFinder that starts with the full partitioned model and consequentially merges partitions (MFP+MERGE option) until the model fit does not increase (*Lanfear et al., 2012*). After the best model has been found, IQ-TREE starts the tree reconstruction under the best model scheme. The phylogenetic analyses were carried out with the *-spp* option that allowed each partition to have its own evolutionary rate. The RAxML-HPC2 V.8.2.10 analysis was carried out on CIPRES using the GTR+CAT option.

Support for nodes was evaluated with 1,000 ultrafast bootstrap (UFBoot2) approximations (*Hoang et al., 2018*) in IQ-TREE, and SH-like approximate likelihood ratio test (*Guindon et al., 2010*). Additionally, we implemented rapid bootstrap (RBS) in RAxML (*Stamatakis, Hoover & Rougemont, 2008*). To reduce the risk of overestimating branch supports in UFBoot2 test, we implemented *-bnni* option, which optimizes each bootstrap tree using a hill-climbing nearest neighbor interchange search. Trees were visualized and edited in FigTree v1.4.3 software (*Rambaut, 2012*). The final trees were rooted with species of the families Sematuridae, Epicopeiidae, Pseudobistonidae and Uraniidae following previous hypotheses proposed in *Regier et al. (2009*, *2013*), *Rajaei et al. (2015)* and *Heikkilä et al. (2015)*.

## Taxonomic decisions

The electronic version of this article in Portable Document Format (PDF) will represent a published work according to the International Commission on Zoological Nomenclature (*International Commission on Zoological Nomenclature, 2012*), and hence the new names contained in the electronic version are effectively published under that Code from the electronic edition alone. This published work and the nomenclatural acts it contains have been registered in ZooBank. The ZooBank LSIDs (Life Science Identifiers) can be resolved and the associated information viewed through any standard web browser by appending the LSID to the prefix http://zoobank.org/. For this publication: LSIDurn:lsid:zoobank.org: pub:662A9A18-B620-45AA-B4B1-326086853316. The online version of this work is archived and available from the following digital repositories: PeerJ, PubMed Central and CLOCKSS.

## RESULTS

### Searching strategies and model selection

The ModelFinder analysis resulted in 26 partitions with associated best-fit models (Table 1). IQ-TREE and RAxML analyses resulted in trees with nearly identical topology. Also, the different methods of evaluating robustness tended to agree in supporting the same nodes. However, in most of the cases UFBoot2 from IQ-TREE showed higher support values compared to RBS in RAxML (RAxML tree with support values is shown in

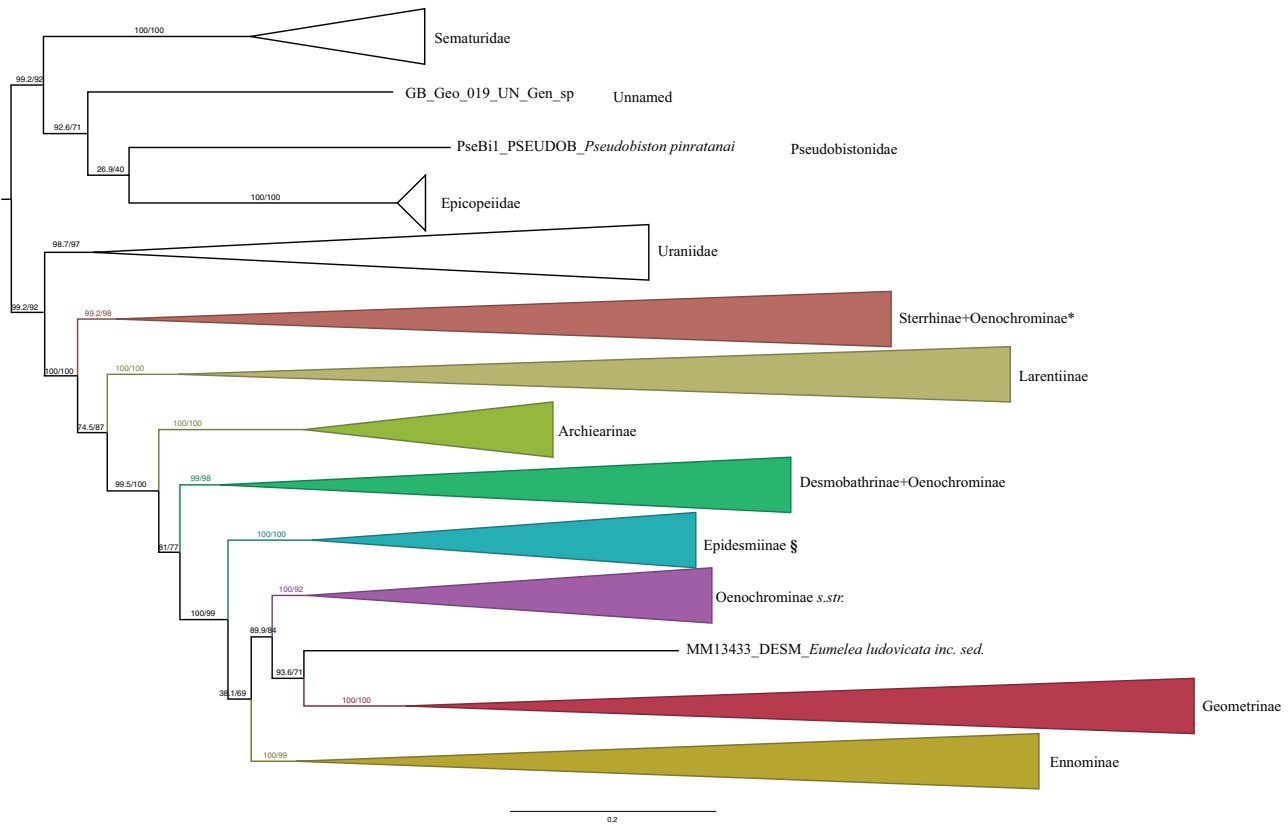

**Figure 1 Evolutionary relationships of major groups of the family Geometridae.** Numbers above branches are SH-aLRT support (%)/ultrafast bootstrap support, UFBoot2(%), for nodes to the right of the numbers. Values of SH ≥ 80 and UFBoot2 ≥ 95 indicate well-supported clades (*Trifinopoulos & Minh, 2018*). *Formal taxonomic treatment will be dealt with in P. Sihvonen et al., 2019, unpublished data. § Epidesmiinae **subfam. nov.** See Oenochrominae section for more details.                              

Data S3). SH-like and UFBoot2 performed similarly, with UFBoot2 showing slightly higher values, and both tended to show high support for the same nodes (Fig. 1). As noted by the authors of IQ-TREE, values of SH ≥ 80 and UFBoot2 ≥ 95 indicate well-supported clades (*Trifinopoulos & Minh, 2018*).

## General patterns in the phylogeny of Geometridae

Analyses of the dataset of 1,206 terminal taxa, comprising up to 11 markers and an alignment length of 7,665 bp recovered topologies with many well-supported clades. About 20 terminal taxa are recovered as very similar genetically and they are likely to represent closely related species, subspecies or specimens of a single species. The examination of their taxonomic status is not the focus of this study, so the number of unique species in the analysis is slightly less than 1,200. Our findings confirm the monophyly of Geometridae (values of SH-like, UFBoot2= 100) (Fig. 1). The general patterns in our phylogenetic hypotheses suggest that Sterrhinae are the sister group to the rest of Geometridae. This subfamily is recovered as monophyletic when three genera traditionally included in Oenochrominae are considered to belong to Sterrhinae

**Table 1 Evolutionary models recovered in ModelFinder.**

| Evolutionary models | Codon position | Data type |
| --- | --- | --- |
| SYM+R5 | ArgK_pos1 | Nuclear |
| SYM+R4 | ArgK_pos2_Ca-ATPase_pos2 | Nuclear |
| GTR+F+R6 | ArgK_pos3 | Nuclear |
| GTR+F+R5 | Ca-ATPase_pos1_IDH_pos1 | Nuclear |
| SYM+I+G4 | Ca-ATPase_pos3 | Nuclear |
| SYM+I+G4 | CAD_pos1 | Nuclear |
| K3P+I+G4 | CAD_pos2 | Nuclear |
| GTR+F+R7 | CAD_pos3 | Nuclear |
| TIM2+F+I+G4 | COI_pos1 | Mitochondrial |
| K2P+R8 | COI_pos2_MDH_pos2_RpS5_pos2_WntGeo_pos2 | Mitochondrial/Nuclear |
| GTR+F+ASC+R10 | COI_pos3 | Mitochondrial |
| TIM2e+R10 | EF1a_pos1 | Nuclear |
| TIM+F+I+G4 | EF1a_pos2 | Nuclear |
| SYM+R10 | EF1a_pos3_GAPDH_pos3_RpS5_pos3 | Nuclear |
| TVM+F+I+G4 | GAPDH_pos1 | Nuclear |
| SYM+I+G4 | GAPDH_pos2 | Nuclear |
| GTR+F+R4 | IDH_pos2 | Nuclear |
| SYM+R6 | IDH_pos3 | Nuclear |
| GTR+F+I+G4 | MDH_pos1 | Nuclear |
| SYM+I+G4 | MDH_pos3 | Nuclear |
| SYM+I+G4 | Nex9_pos1 | Nuclear |
| K3P+I+G4 | Nex9_pos2 | Nuclear |
| GTR+F+R6 | Nex9_pos3 | Nuclear |
| SYM+I+G4 | RpS5_pos1 | Nuclear |
| GTR+F+I+G4 | WntGeo_pos1 | Nuclear |
| SYM+R7 | WntGeo_pos3 | Nuclear |

(see details below). Tribes in Sterrhinae, such as Timandriini, Rhodometrini, Lythriini, Rhodostrophiini and Cyllopodini, are not recovered as monophyletic (Fig. 2). A detailed analysis, including formal changes to the classification of Sterrhinae, will be provided by P. Sihvonen et al., 2019, unpublished data.

The monophyly of Larentiinae is established in previous studies (*Sihvonen et al., 2011*; *Õunap, Viidalepp & Truuverk, 2016*) and our results are largely in agreement with their hypotheses. However, our results do not support the sister relationship between Sterrhinae and Larentiinae found in previous studies. Rather, we find that Sterrhinae are the sister to the rest of Geometridae. Within Larentiinae, in concordance with recent findings (*Sihvonen et al., 2011*; *Õunap, Viidalepp & Truuverk, 2016*; *Strutzenberger et al., 2017*), we find Dyspteridini as the sister group to the remaining Larentiinae (Fig. 3). Phylogenetic relationships within Larentiinae were treated in detail by *Õunap, Viidalepp & Truuverk (2016)*. Further details of the analyses and changes to the classification of Larentiinae will be discussed by G. Brehm et al., 2019, unpublished data and E. Õunap et al., 2019, unpublished data.

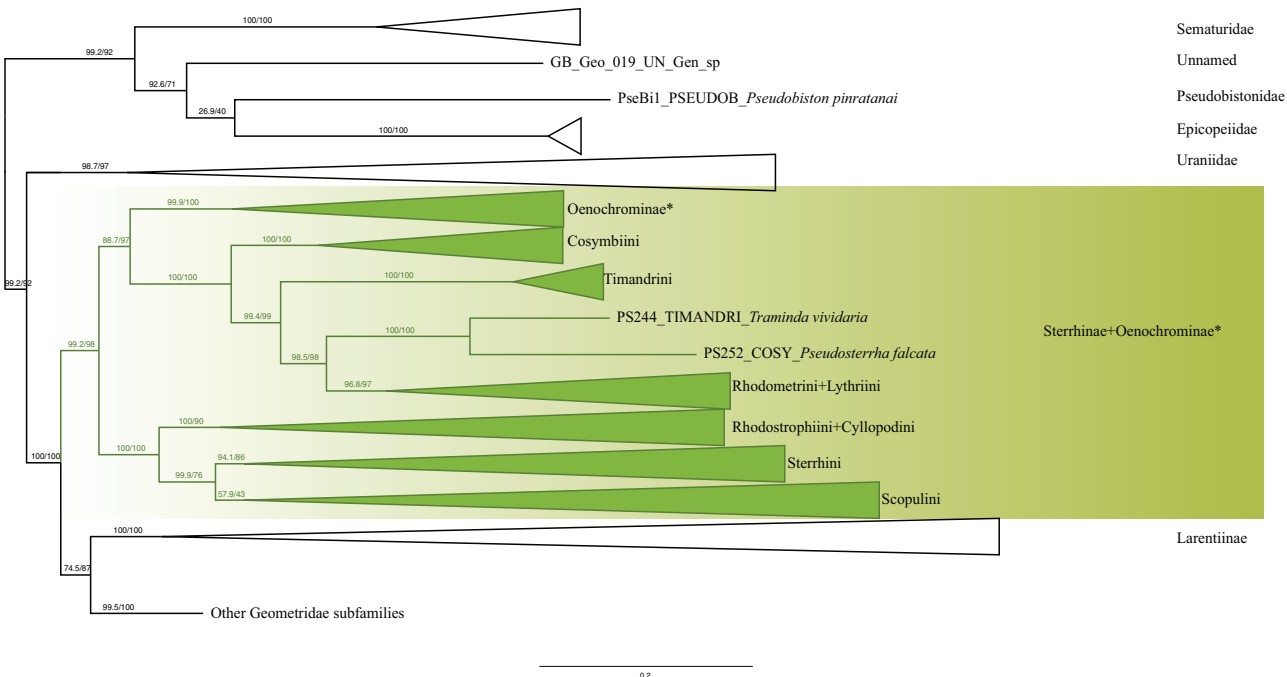

**Figure 2 Evolutionary relationships of the subfamily Sterrhinae.** Numbers above branches are SH-aLRT support (%)/ultrafast bootstrap support, UFBoot2(%), for nodes to the right of the numbers. Values of SH ≥ 80 and UFBoot2 ≥ 95 indicate well-supported clades (*Trifinopoulos & Minh, 2018*). *Formal taxonomic treatment will be dealt with in P. Sihvonen et al., 2019, unpublished data.

Archiearinae are represented by more taxa than in a previous study (*Sihvonen et al., 2011*). Archiearinae grouped as sister to Oenochrominae + Desmobathrinae complex + *Eumelea* + Geometrinae and Ennominae (Fig. 4). The monophyly of this subfamily is well supported (values of SH-like, UFBoot2 = 100). However, as in the previous study (*Sihvonen et al., 2011*), the Australian genera *Dirce* Prout, 1910 and *Acalyphes* Turner, 1926 are not part of Archiearinae but can clearly be assigned to Ennominae. Unlike previously assumed (e.g., *McQuillan & Edwards, 1996*), the subfamily Archiearinae probably does not occur in Australia, despite superficial similarities of *Dirce*, *Acalyphes* and Archiearinae.

Desmobathrinae were shown to be paraphyletic by *Sihvonen et al. (2011)*. In our analysis, the monophyly of this subfamily is not recovered either, as we find two genera traditionally placed in Oenochrominae (i.e. *Zanclopteryx* Herrich-Schäffer, (1855) and *Racasta* Walker, 1861) nested within Desmobathrinae (Fig. 4). We formally transfer these genera to Desmobathrinae. In the revised sense, Desmobathrinae form a well-supported group with two main lineages. One of them comprises *Ozola* Walker, 1861, *Derambila* Walker, 1863 and *Zanclopteryx*. This lineage is sister to a well-supported clade comprising *Conolophia* Warren, 1894, *Noreia* Walker, 1861, *Leptoctenopsis* Warren, 1897, *Racasta*, *Ophiogramma* Hübner, 1831, *Pycnoneura* Warren, 1894 and *Dolichoneura* Warren, 1894.

Oenochrominae in the broad sense are not a monophyletic group. However, Oenochrominae *sensu stricto* (*Scoble & Edwards, 1990*) form a well-supported lineage

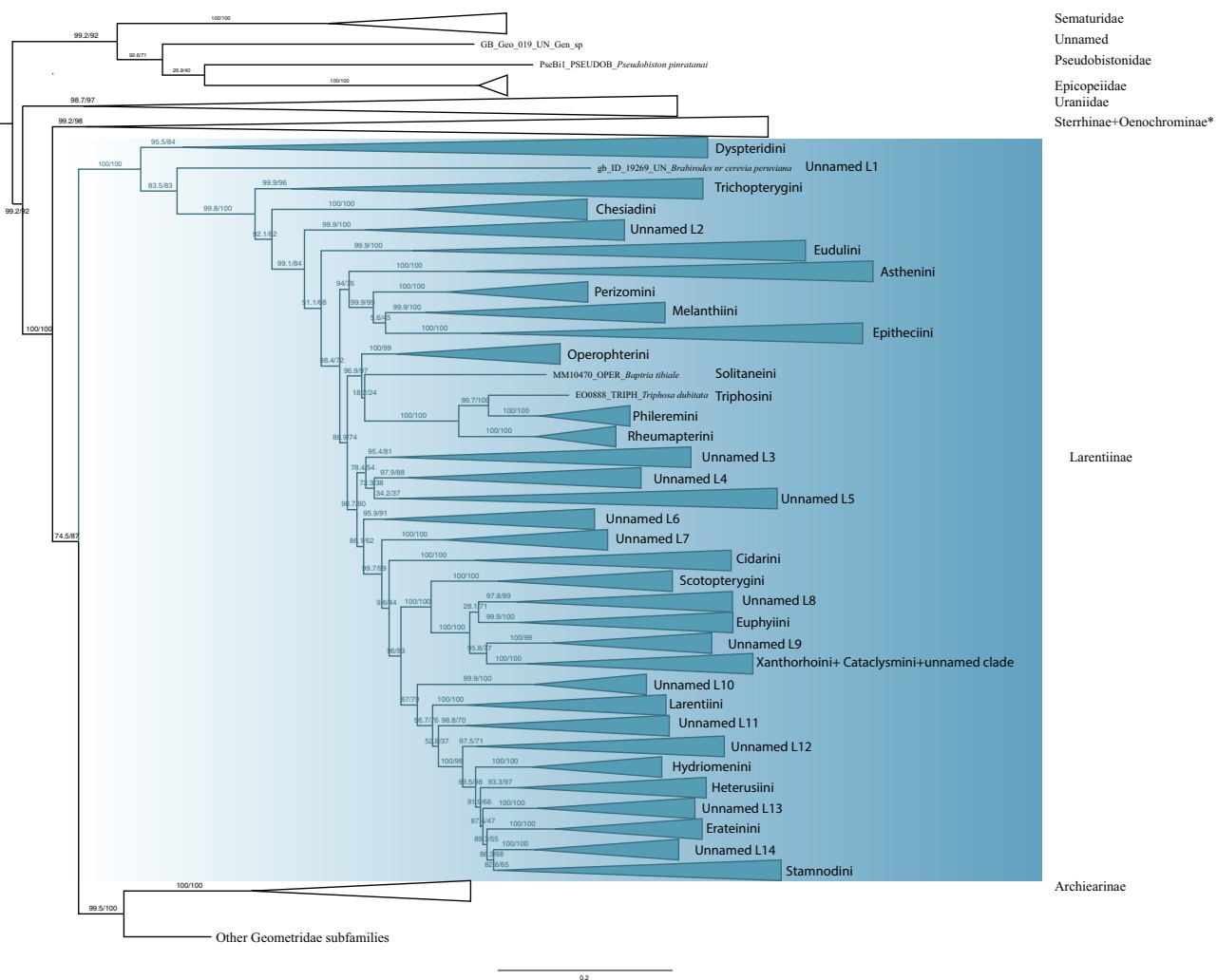

**Figure 3 Evolutionary relationships of the subfamily Larentiinae.** Numbers above branches are SH-aLRT support (%)/ultrafast bootstrap support, UFBoot2(%), for nodes to the right of the numbers. Values of SH ≥ 80 and UFBoot2 ≥ 95 indicate well-supported clades (*Trifinopoulos & Minh, 2018*). *Formal taxonomic treatment will be dealt with in P. Sihvonen et al., 2019, unpublished data.

comprising two clades. One of them contains a polyphyletic *Oenochroma* with *Oenochroma infantilis* Prout, 1910 being sister to *Dinophalus* Prout, 1910, *Hypographa* Guenée, 1858, *Lissomma* Warren, 1905, *Sarcinodes* Guenée, 1858 and two further species of *Oenochroma*, including the type species *Oenochroma vinaria* Guenée, 1858. The other clade comprises *Monoctenia* Guenée, 1858, *Onycodes* Guenée, 1858, *Parepisparis* Bethune-Baker, 1906, *Antictenia* Prout, 1910, *Arthodia* Guenée, 1858, *Gastrophora* Guenée, 1858 and *Homospora* Turner, 1904 (Fig. 4). Most of the remaining genera traditionally placed in Oenochrominae, including e.g. *Epidesmia* Duncan & Westwood, 1841, form a well-supported monophyletic clade that is sister to Oenochrominae *s.str.* + *Eumelea ludovicata* + Geometrinae + Ennominae assemblage.

The genus *Eumelea* Duncan & Westwood, 1841 has an unclear phylogenetic position in our analyses. The IQ-TREE result suggests *Eumelea* to be sister to the subfamily
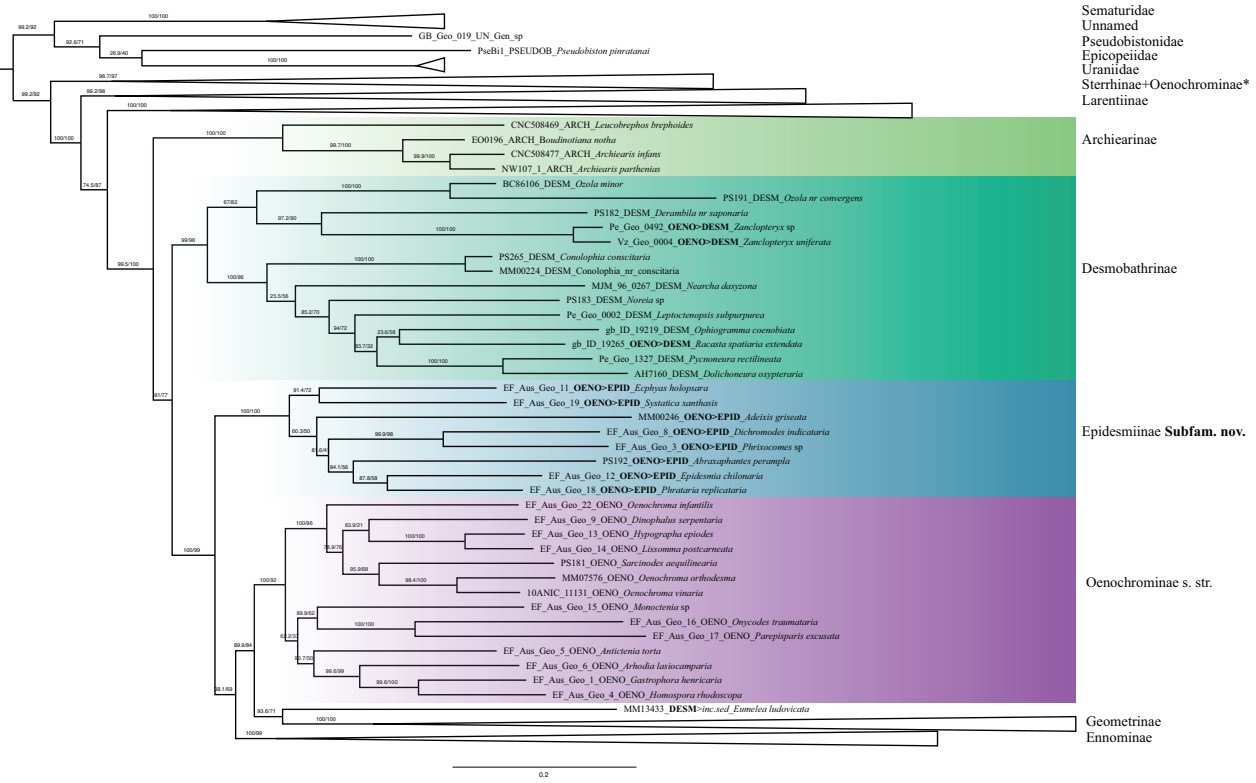

**Figure 4 Phylogenetic relationships of the subfamilies Archierinae, Desmobathrinae, Epidesmiinae subfam. nov., Oenochrominae.** Numbers above branches are SH-aLRT support (%)/ultrafast bootstrap support, UFBoot2(%), for nodes to the right of the numbers. Values of SH ≥ 80 and UFBoot2 ≥ 95 indicate well-supported clades (*Trifinopoulos & Minh, 2018*). Taxonomic changes are indicated by a symbolized arrow >. *Formal taxonomic treatment will be dealt with in P. Sihvonen et al., 2019, unpublished data.

Geometrinae (SH-like = 93.6, UFBoot2 = 71), whereas RAxML recovered *Eumelea* in Ennominae as sister of *Plutodes* Guenée, 1858 (RBS = 60).

The monophyly of Geometrinae is well supported (Fig. 5) and in IQ-TREE results Geometrinae are recovered as the sister-taxon of *Eumelea*. The *Eumelea* + Geometrinae clade is sister to Oenochrominae *s.str*. Although a recent phylogenetic study proposed several taxonomic changes (*Ban et al., 2018*), the tribal composition in Geometrinae is still problematic. Many tribes are recovered as paraphyletic. Our results suggest that *Ornithospila* Warren, 1894 and *Agathia* Guenée, 1858 form a lineage sister to the rest of Geometrinae. *Chlorodontopera* is placed as an isolated lineage sister to Aracimini, Neohipparchini, Timandromorphini, Geometrini and Comibaenini, which are recovered as monophyletic groups, respectively. Synchlorini are nested within Nemoriini in a well-supported clade (support branch SH-like = 98.3, UFBoot2 = 91, RBS = 93). The monophyly of Pseudoterpnini could not be recovered, instead this tribe splits up into three well-defined groups. Several genera currently placed in Pseudoterpnini *s.l.* are recovered as an independent lineage clearly separate from Pseudoterpnini *s.str*. (SH-like, UFBoot2 = 100). *Xenozancla* Warren, 1893 is sister to a clade comprising Dysphaniini and Pseudoterpnini *s.str*. Hemitheini *sensu Ban et al. (2018)* are recovered as a well-supported

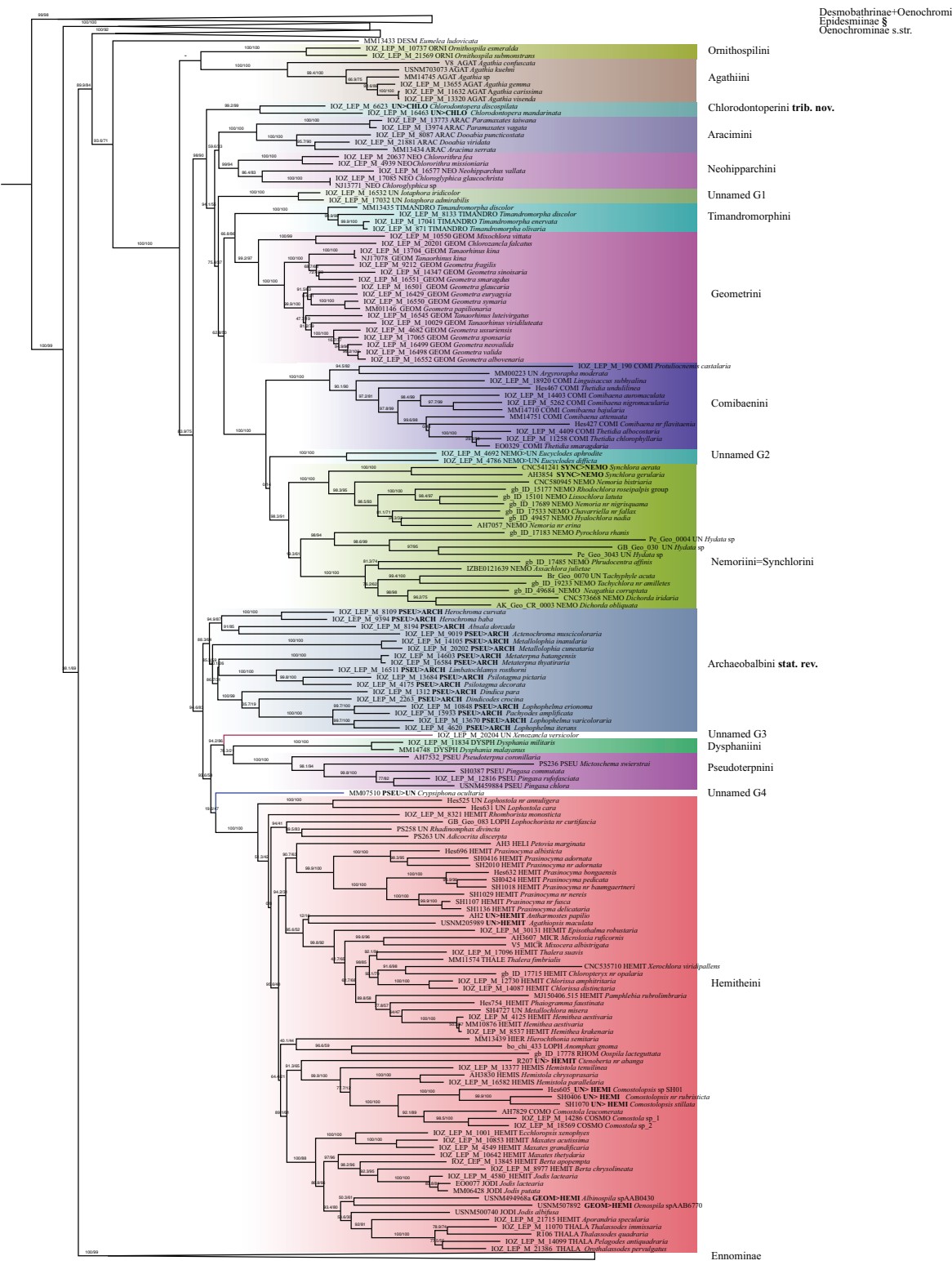

**Figure 5 Evolutionary relationships of the subfamily Geometrinae.** Numbers above branches are SH-aLRT support (%)/ultrafast bootstrap support, UFBoot2(%), for nodes to the right of the numbers. Values of SH ≥ 80 and UFBoot2 ≥ 95 indicate well-supported clades (*Trifinopoulos & Minh, 2018*). Taxonomic changes are indicated by a symbolized arrow >. § New subfamily.

clade. *Crypsiphona ocultaria* Meyrick, 1888 was resolved as a single lineage, close to *Lophostola* + Hemitheini .

Ennominae are strongly supported as monophyletic in IQ-TREE analyses (SH-like = 100, UFBoot2 = 99) whereas in RAxML the monophyly is weakly supported (RBS = 63). Detailed results concerning the classification, especially for the Neotropical taxa, will be presented by G. Brehm et al. (2019, unpublished data), but the main results are summarized here (Fig. 6). Very few tribes are monophyletic according to the results of the present study. One group of Neotropical taxa currently assigned to Gonodontini (unnamed E1), *Idialcis* Warren, 1906 (unnamed clade E2), Gonodontini *s.str.*, Gnophini, Odontoperini, unnamed clade E3, Nacophorini and Ennomini (*sensu Beljaev, 2008*) group together (SH-like = 90.3, UFBoot2 = 87). Ennomini were sister to this entire group. Campaeini is recovered as sister of Alsophilini + Wilemaniini and Colotoini. In turn they are sister to a clade comprising a number of taxa. These include the New Zealand genus *Declana* Walker, 1858 (unnamed E4) which appear as sister to a large complex including *Acalyphes* Turner, 1926 + *Dirce* Prout, 1910, Lithinini, intermixed with some genera currently placed in Nacophorini and Diptychini.

*Neobapta* Warren, 1904 and *Oenoptila* Warren, 1895 form an independent lineage (unnamed E5) sister to Theriini, which in turn form a supported clade with *Lomographa* (Baptini) (SH-like, UFBoot2 = 100). Likewise, we recovered *Erastria* Hübner, 1813 + *Metarranthis* Warren, 1894 (both as unnamed E5) as sister to Plutodini + Palyadini. The IQ-TREE analyses show Palyadini as a well-defined lineage, sister to *Plutodes*. However, in RAxML analyses, *Eumelea* and *Plutodes* group together and Palyadini cluster with a group of Caberini species. In the IQ-TREE analysis Apeirini formed a lineage with Hypochrosini, Epionini, *Sericosema* Warren, 1895 and *Ithysia* Hübner, 1825. This lineage is in turn sister of African *Drepanogynis* Guenée, 1858 which groups together with *Sphingomima* Warren, 1899, *Thenopa* Walker, 1855 and *Hebdomophruda* Warren, 1897. Caberini are sister to an unnamed clade composed of *Trotogonia* Warren, 1905, *Acrotomodes* Warren, 1895, *Acrotomia* Herrich-Schäffer 1855 and *Pyrinia* Hübner, 1818. Finally, our analyses recover a very large, well-supported clade comprising the tribes Macariini, Cassymini, Abraxini, Eutoeini and Boarmiini (SH-like = 100, UFBoot2 = 99). This large clade has previously been referred to informally as the "boarmiines" by *Forbes (1948)* and *Wahlberg et al. (2010)*. The tribe Cassymini is clearly paraphyletic: genera such as *Cirrhosoma* Warren, 1905, *Berberodes* Guenée, 1858, *Hemiphricta* Warren, 1906 and *Ballantiophora* Butler, 1881 currently included in Cassymini, cluster in their own clade together with *Dorsifulcrum* Herbulot, 1979 and *Odontognophos* Wehrli, 1951. We were unable to include Orthostixinae in the analyses, so we could not clarify the taxonomic position of this subfamily with regard to its possible synonymy with Ennominae (*Sihvonen et al., 2011*).

## DISCUSSION

### Optimal partitioning scheme and support values

The greedy algorithm implemented in ModelFinder to select the best-fitting partitioning scheme combined the codon partitions into 26 subsets (Table 2). These results are not different from previous studies that tested the performance of different data partitioning

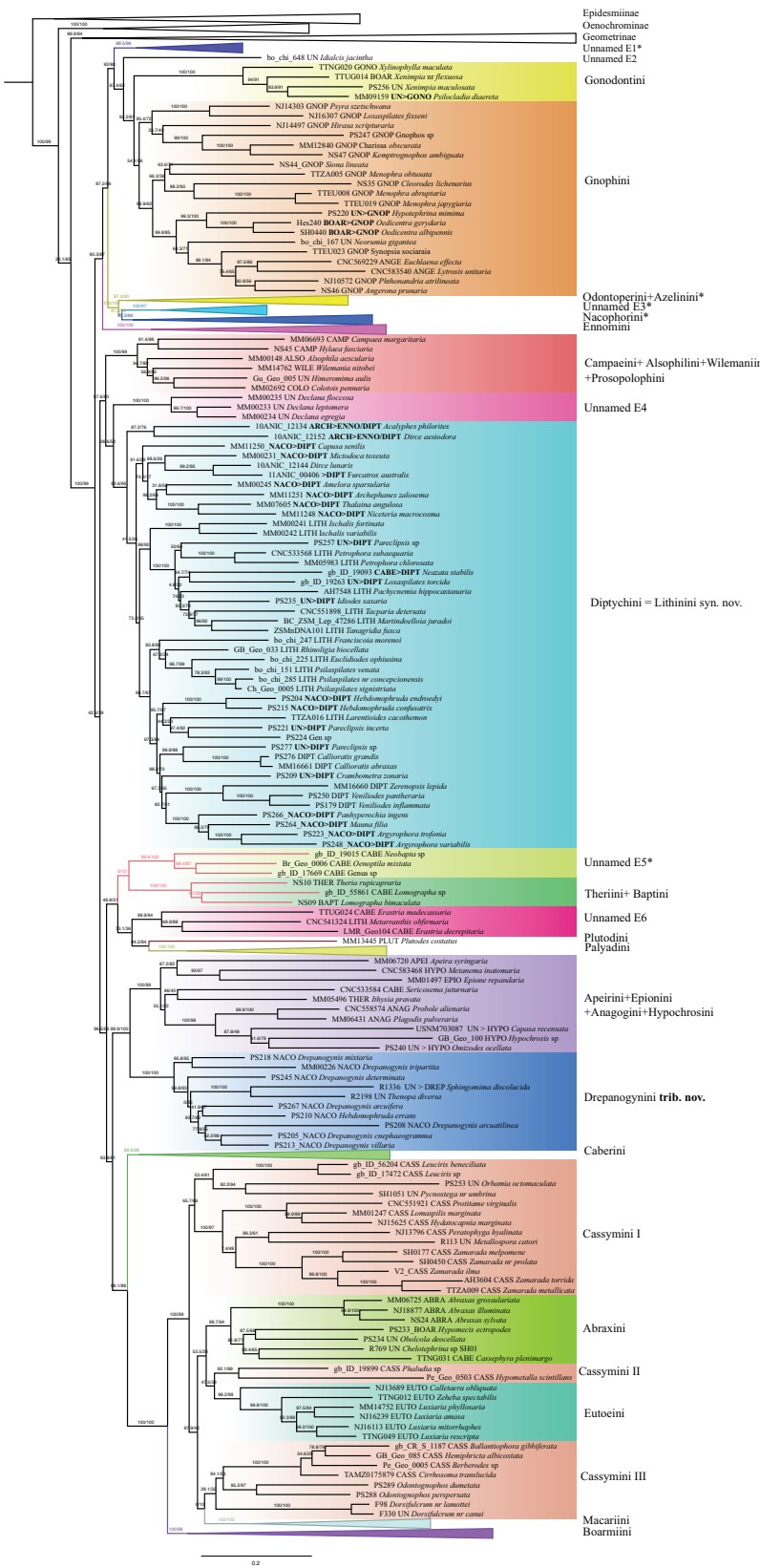

**Figure 6 Evolutionary relationships of the subfamily Ennominae.** Numbers above branches are SH-aLRT support (%)/ultrafast bootstrap support, UFBoot2(%), for nodes to the right of the numbers. Values of SH ≥ 80 and UFBoot2 ≥ 95 indicate well-supported clades (*Trifinopoulos & Minh, 2018*). Taxonomic changes are indicated by a symobolized arrow >. *Formal taxonomic treatment will be dealt with in G. Brehm et al., 2019, unpublished data.

schemes and found that in some cases partitioning by gene can result in suboptimal partitioning schemes and may limit the accuracy of phylogenetic analyses (*Rota, 2011*; *Lanfear et al., 2012*). However, we note that although the AIC and BIC values were lower when the data were partitioned by gene, the tree topology recovered was nevertheless almost the same as when data were partitioned by codon position, suggesting that much of the phylogenetic signal in the data is robust to partitioning schemes. As would be expected, the analyses resulted in some disagreements between the different measures of node support. Ultrafast bootstrap gave the highest support values, followed by SH-like and finally standard bootstrap as implemented in RAxML gave the lowest. Although support indices obtained by these methods are not directly comparable, differences in node support of some clades can be attributed to the small number of markers, insufficient phylogenetic signal or saturated divergence levels (*Guindon et al., 2010*).

## Current understanding of Geometridae phylogeny and taxonomic implications

### Geometridae Leach, 1815

The phylogenetic hypothesis presented in this study is by far the most comprehensive to date in terms of the number of markers, sampled taxa and geographical coverage. In total, our sample includes 814 genera, thus representing 41% of the currently recognized Geometridae genera (*Scoble & Hausmann, 2007*). Previous phylogenetic hypotheses were based mainly on the European fauna and many clades were ambiguously supported due to low taxon sampling. The general patterns of the phylogenetic relationships among the subfamilies recovered in our study largely agrees with previous hypotheses based on morphological characters and different sets of molecular markers (*Holloway, 1997*; *Abraham et al., 2001*; *Yamamoto & Sota, 2007*; *Sihvonen et al., 2011*). However, the results of our larger dataset differ in many details and shed light on the phylogenetic relationships of several, poorly resolved, small subfamilies.

Sterrhinae are recovered as the sister subfamily to the remaining Geometridae. This result is not in concordance with *Sihvonen et al. (2011)*, *Yamamoto & Sota (2007)* and *Regier et al. (2009)*, who found a sister group relationship between Sterrhinae and Larentiinae which in turn were sister to the rest of Geometridae. *Sihvonen et al. (2011)* showed the Sterrhinae + Larentiinae sister relationship with low support, while *Yamamoto & Sota (2007)* and *Regier et al. (2009)* included only a few samples in their analyses. Our analyses include representatives from almost all known tribes currently included in Sterrhinae and Larentiinae. The higher number of markers, improved methods of analysis, the broader taxon sampling as well as the stability of our results suggests that Sterrhinae are indeed the sister group to the remaining Geometridae. Sterrhinae (after transfer of *Ergavia*, *Ametris* and *Macrotes*, see details below), Larentiinae, Archiearinae,

**Table 2  Summary of formally proposed taxonomic changes.**

**Transfer from Archiearinae to Ennominae**

*Acalyphes* Turner, 1926, to Ennominae: Diptychini
*Dirce* Prout, 1910, to Ennominae: Diptychini

**Transfer from Oenochrominae to Desmobathrinae (Desmobathrini):**

*Nearcha* Guest, 1887
*Racasta* Walker, 1861
*Zanclopteryx* Herrich-Schäffer, 1855

**Transfer from Oenochrominae to Epidesmiinae:**

*Abraxaphantes* Warren, 1894
*Adeixis* Warren 1987
*Dichromodes* Guenée 1858
*Ecphyas* Turner, 1929
*Epidesmia* Duncan & Westwood, 1841
*Phrixocomes* Turner, 1930
*Phrataria* Walker, 1863
*Systatica* Turner, 1904

**New tribe combinations in Ennominae**

*Psilocladia* Warren, 1898, from unassigned to Gonodontini

*Oedicentra* Warren, 1902, from Boarmiini to Gnophini

*Hypotephrina* Janse, 1932, from unassigned to Gnophini

*Capusa* Walker, 1857, from Nacophorini to Diptychini

*Mictodoca* Meyrick, 1892, from Nacophorini to Diptychini

*Furcatrox* McQuillan, 1996, from Nacophorini to Diptychini

*Amelora* Guest, 1897, from Nacophorini to Diptychini

*Archephanes* Turner, 1926, from Nacophorini to Diptychini

*Thalaina* Walker, 1855, from Nacophorini to Diptychini

*Niceteria* Turner, 1929, from Nacophorini to Diptychini

*Neazata* Warren, 1906 from Caberini to Diptychini

*Idiodes* Guenée, 1858 from unassigned to Diptychini

*Panhyperochia* Krüger, 2013, from Nacophorini to Diptychini

*Mauna* Walker, 1865, from Nacophorini to Diptychini

*Pareclipsis* Warren, 1894, from unassigned to Diptychini

*Crambometra* Prout, 1915, from unassigned to Diptychini

*Hebdomophruda* Warren, 1897, from Nacophorini to Diptychini

*Pareclipsis* Warren, 1894, from unassigned to Diptychini

*Capasa* Walker 1866, from unassigned to Hypochrosini

*Omizodes* Warren, 1894, from unassigned to Hypochrosini

*Metallospora* Warren, 1905, from unassigned to Cassymini

*Obolcola* Walker, 1862, from unassigned to Abraxini

*Chelotephrina* Fletcher, 1958 from unassigned to Abraxini

*Cassephyra* Holloway, 1994 from Cassymini to Abraxini

*Thenopa* Walker, 1855 from unassigned to Drepanogynini

*Drepanogynis* Guenée, 1858 from Nacophorini to Drepanogynini

| New tribe combinations in Geometrinae | |
|---|---|
| *Agathiopsis* Warren 1896, from unassigned to Hemitheini | |
| *Albinospila* Holloway, 1996, from Geometrini to Hemitheini | |
| *Antharmostes* Warren 1899, from unassigned to Hemitheini | |
| *Ctenoberta* Prout 1915, from unassigned to Hemitheini | |
| *Comostolopsis* Warren 1902, from unassigned to Hemitheini | |
| *Oenospila* Swinhoe 1892, from Geometrini to Hemitheini | |
| **New and upgraded tribes in Geometrinae** | **Included taxa** |
| Archaeobalbini, **stat. rev.** | Type genus: *Herochroma* Swinhoe, 1893 (syn. *Archaeobalbis* Prout, 1912). Other included genera: *Pachyodes* Guenée, 1858; *Metallolophia* Warren, 1895; *Actenochroma* Warren, 1893; *Absala* Swinhoe 1893; *Metaterpna* Yazaki, 1992; *Limbatochlamys* Rothschild, 1894; *Psilotagma* Warren, 1894; *Dindica* Warren, 1893; *Dindicodes* Prout, 1912; *Lophophelma* Prout, 1912. |
| Chlorodontoperini, Murillo-Ramos, Sihvonen & Brehm, **trib. nov.** | Type genus: *Chlorodontopera* Warren, 1893. Species included: *C. discospilata* (Moore, 1867); *C. mandarinata* (Leech, 1889); *C. chalybeata* (Moore, 1872); *C. taiwana* (Wileman, 1911). |
| **New tribe in Ennominae** | **Included taxa** |
| Drepanogynini, Murillo-Ramos, Sihvonen & Brehm, **trib. nov.** | Type genus: *Drepanogynis* Guenée, 1858. Other included genera: *Thenopa* Walker, 1855. Species included, genus combination uncertain (*incertae sedis*): *"Sphingomima" discolucida* Herbulot, 1995 (transferred from unassigned to Drepanogynini); *"Hebdomophruda" errans* Prout, 1917 (transferred from Nacophorini to Drepanogynini). |
| **Synonymized tribes** | **Valid tribe** |
| Lithinini Forbes, 1948, **syn. nov.** | Diptychini Janse, 1933 (Ennominae) |
| Synchlorini Ferguson, 1969 **syn. nov.** | Nemoriini Gumppenberg, 1887 (Geometrinae) |
| **Incertae sedis** | |
| *Eumelea* Duncan & Westwood, 1841 | |

Geometrinae and Ennominae were highly supported as monophyletic. Oenochrominae and Desmobathrinae formed polyphyletic and paraphyletic assemblages, respectively. The monophylies of Oenochrominae and Desmobathrinae have long been questioned. Morphological studies addressing Oenochrominae or Desmobathrinae have been limited and the majority of genera have never been examined in depth. In addition, it has been very difficult to establish the boundaries of these subfamilies on the basis of morphological structures (*Scoble & Edwards, 1990*). *Sihvonen et al. (2011)* showed that neither Oenochrominae nor Desmobathrinae were monophyletic, but these results were considered preliminary due to the limited number of sampled taxa, and as a consequence no formal transfers of taxa were proposed.

The systematic status of Orthostixinae remains uncertain because it was not included in our study. *Sihvonen et al. (2011)* included the genus *Naxa* Walker, 1856, formally placed in Orthostixinae, and found it to be nested within Ennominae. However, only three genes were successfully sequenced from this taxon, and its position in the phylogenetic tree turned out to be highly unstable in our analyses. It was thus excluded from our dataset.

*Orthostixis* Hübner, 1823, the type genus of the subfamily, needs to be included in future analyses.

**Sterrhinae Meyrick, 1892**

We included 74 Sterrhinae taxa in our analyses, with all tribes recognized in *Forum Herbulot (2007)* being represented. The recovered patterns generally agree with previous phylogenetic hypotheses of the subfamily (*Sihvonen & Kaila, 2004*, *Sihvonen et al., 2011*). The genera *Ergavia* Walker, 1866, *Ametris* Guenée, 1858 and *Macrotes* Westwood, 1841, which currently are placed in Oenochrominae were found to form a well-defined lineage within Sterrhinae with strong support (SH-Like = 99 UFBoot2 = 100). These genera are distributed in the New World, whereas the range of true Oenochrominae is restricted to the Australian and Oriental Regions. *Sihvonen et al. (2011)* already found that *Ergavia* and *Afrophyla* Warren, 1895 belong to Sterrhinae and suggested more extensive analyses to clarify the position of these genera, which we did. *Afrophyla* was transferred to Sterrhinae by *Sihvonen & Staude (2011)* and *Ergavia*, *Ametris* and *Macrotes* (plus *Almodes* Guenée, (1858)) will be transferred by P. Sihvonen et al. (2019, unpublished data).

Cosymbiini, Timandrini, Rhodometrini and Lythriini are closely related as shown previously (*Sihvonen & Kaila, 2004*; *Õunap, Viidalepp & Saarma, 2008*; *Sihvonen et al., 2011*). Cosymbiini appear as sister to the Timandrini + *Traminda* Saalmüller, 1891 + *Pseudosterrha* Warren, 1888 and Rhodometrini + Lythriini clade. Lythriini are closely related to Rhodometrini as shown by *Õunap, Viidalepp & Saarma (2008)* with both molecular and morphological data. *Traminda* (Timandrini) and *Pseudosterrha* (Cosymbiini) grouped together forming a lineage that is sister to the Rhodometrini + Lythriini clade (Fig. 2).

Rhodostrophiini and Cyllopodini were recovered as polyphyletic with species of Cyllopodini clustering within Rhodostrophiini. Similar results were recovered previously (*Sihvonen & Kaila, 2004*; *Sihvonen et al., 2011*), suggesting that additional work is needed to be done to clarify the status and systematic positions of these tribes. Sterrhini and Scopulini were recovered as sister taxa as proposed by *Sihvonen & Kaila (2004)*, *Hausmann (2004)*, *Õunap, Viidalepp & Saarma (2008)* and *Sihvonen et al. (2011)*. Our new phylogenetic hypothesis constitutes a large step towards understanding the evolutionary relationships of the major lineages of Sterrhinae. Further taxonomic changes and more detailed interpretation of the clades will be dealt with by P. Sihvonen et al. (2019, unpublished data).

**Larentiinae Duponchel, 1845**

Larentiinae are a monophyletic entity (Fig. 3). In concordance with the results of *Sihvonen et al. (2011)*, *Viidalepp (2011)*, *Õunap, Viidalepp & Truuverk (2016)* and *Strutzenberger et al. (2017)*, Dyspteridini are supported as sister to all other larentiines. Remarkably, *Brabirodes* Warren, 1904 forms an independent lineage. Chesiadini are monophyletic and sister to all larentiines except Dyspteridini, *Brabirodes* and Trichopterygini. These results do not support the suggestion by *Viidalepp (2006)* and *Sihvonen et al. (2011)* that Chesiadini are sister to Trichopterygini.

In our phylogenetic hypothesis, Asthenini are sister to the Perizomini + Melanthiini + Eupitheciini clade. These results do not fully agree with *Õunap, Viidalepp & Truuverk (2016)* who found Asthenini to be sister to all Larentiinae except Dyspteridini, Chesiadini, Trichopterygini and Eudulini. However, our results do support the Melanthiini + Eupitheciini complex as a sister lineage to Perizomini. *Sihvonen et al. (2011)* recovered Phileremini and Rheumapterini as well-supported sister taxa. Our results suggest *Triphosa dubitata* Linnaeus 1758 (Triphosini) is sister to Phileremini, with Rheumapterini sister to this clade. Cidariini were recovered as paraphyletic, as the genera *Coenotephria* Prout, 1914 and *Lampropteryx* Stephens, 1831 cluster in a different clade (unnamed clade L7) apart from the lineage comprising the type genus of the tribe, *Cidaria* Treitschke, 1825. *Ceratodalia* Packard, 1876, currently placed in Hydriomenini and *Trichodezia* Warren, 1895 are nested within Cidariini. These results are not in concordance with *Õunap, Viidalepp & Truuverk (2016)*, who regarded this tribe to be monophyletic. Scotopterygini are sister to a lineage comprising *Ptychorrhoe blosyrata* Guenée (1858), *Disclisioprocta natalata* (Walker, 1862) (placed in the unnamed clade L8), Euphyiini, an unnamed clade L9 comprising the genera *Pterocypha*, *Archirhoe* and *Obila*, Xanthorhoini and Cataclysmini. Euphyiini are monophyletic, but Xanthorhoini are recovered as mixed with Cataclysmini. The same findings were shown by *Õunap, Viidalepp & Truuverk (2016)*, but no taxonomic rearrangements were proposed. Larentiini are monophyletic and sister of Hydriomenini, Heterusiini, Erateinini, Stamnodini and some unnamed clades (L11–14). Although with some differences, our results support the major phylogenetic patterns of *Õunap, Viidalepp & Truuverk (2016)*.

Despite substantial progress, the tribal classification and phylogenetic relationships of Larentiinae are far from being resolved (*Õunap, Viidalepp & Truuverk, 2016*). *Forbes (1948)* proposed eight tribes based on morphological information, *Viidalepp (2011)* raised the number to 23 and *Õunap, Viidalepp & Truuverk (2016)* recovered 25 tribes studying 58 genera. Our study includes 23 of the currently recognized tribes and 125 genera (with an emphasis on Neotropical taxa). However, the phylogenetic position of many taxa remains unclear, and some tropical genera have not yet been formally assigned to any tribe. Formal descriptions of these groups will be treated in detail by G. Brehm et al. (2019, unpublished data) and E. Õunap et al. (2019, unpublished data).

### Archiearinae Fletcher, 1953

The hypothesis presented in this study recovered Archiearinae as a monophyletic entity after some taxonomic rearrangements are performed. This subfamily was previously considered as sister to Geometrinae + Ennominae (*Abraham et al., 2001*), whereas *Yamamoto & Sota (2007)* proposed them to be the sister-taxon to Orthostixinae + Desmobathrinae. Our findings agree with *Sihvonen et al. (2011)* who recovered Archiearinae as the sister-taxon to the rest of Geometridae excluding Sterrhinae and Larentiinae, although only one species was included in their study. *Archiearis* Hübner, (1823) is sister to *Boudinotiana* Esper, 1787 and these taxa in turn are sister to *Leucobrephos* Grote, 1874 (Fig. 4). The southern hemisphere Archiearinae require more attention. *Young (2006)* suggested that two Australian Archiearinae genera, *Dirce* and

*Acalyphes*, actually belong to Ennominae. Our analyses clearly support this view and we therefore propose to formally transfer *Dirce* and *Acalyphes* to Ennominae (all formal taxonomic changes are provided in Table 2). Unfortunately, the South American Archiearinae genera *Archiearides* Fletcher, 1953 and *Lachnocephala* Fletcher, 1953, and Mexican *Caenosynteles* Dyar, 1912 (*Pitkin & Jenkins, 2004*), could not be included in our analyses. These presumably diurnal taxa may only be superficially similar to northern hemisphere Archiearinae as was the case with Australian *Dirce* and *Acalyphes*.

**Desmobathrinae Meyrick, 1886**

Taxa placed in Desmobathrinae were formerly recognized as Oenochrominae genera with slender appendages. *Holloway (1996)* revived Desmobathrinae from synonymy with Oenochrominae and divided it into the tribes Eumeleini and Desmobathrini. Desmobathrinae species have a pantropical distribution and they apparently (still) lack recognized morphological apomorphies (*Holloway, 1996*). Our phylogenetic analysis has questioned the monophyly of Desmobathrinae *sensu* Holloway because some species currently placed in Oenochrominae were embedded within the group (see also *Sihvonen et al., 2011*), and also the phylogenetic position of the tribe Eumeleini is unstable (see below). Desmobathrinae can be regarded as a monophyletic group after the transfer of *Zanclopteryx, Nearcha* and *Racasta* from Oenochrominae to Desmobathrinae, and the removal of Eumeleini (Table 2). Desmobathrinae as circumscribed here are an independent lineage that is sister to all Geometridae except Sterrhinae, Larentiinae and Archiearinae.

The monobasic Eumeleini has had a dynamic taxonomic history: *Eumelea* was transferred from Oenochrominae *s.l.* to Desmobathrinae based on the pupal cremaster (*Holloway, 1996*), whereas *Beljaev (2008)* pointed out that *Eumelea* could be a member of Geometrinae based on the skeleto-muscular structure of the male genitalia. Molecular studies (*Sihvonen et al., 2011*, *Ban et al., 2018*) suggested that *Eumelea* was part of Oenochrominae *s.str.*, but these findings were not well-supported and no formal taxonomic changes were proposed. Our analyses with IQTREE and RAxML recovered Eumeleini in two very different positions, either as sister to Geometrinae (SH-like = 93.6, UFBoot2 = 71) (Figs. 4 and 5), or as sister of *Plutodes* in Ennominae (RBS = 60) (Data S3). The examination of morphological details suggests that the position as sister to Geometrinae is more plausible: hindwing vein M2 is present and tubular; anal margin of the hindwing is elongated; and large coremata originate from the saccus (*Holloway, 1994*, our observations). The morphology of *Eumelea* is partly unusual, and for that reason we illustrate selected structures (Data S4), which include for instance the following: antennae and legs of both sexes are very long; forewing vein Sc (homology unclear) reaches wing margin; in male genitalia coremata are extremely large and branched; uncus is cross-shaped (cruciform); tegumen is narrow and it extends ventrally beyond the point of articulation with vinculum; saccus arms are extremely long, looped; and vesica is with lateral rows of cornuti. However, the green geoverdin pigment concentration of *Eumelea* is low in comparison to Geometrinae (*Cook et al., 1994*). We tentatively conclude that *Eumelea* is probably indeed associated with Geometrinae. However, since eleven genetic

markers were not sufficient to clarify the phylogenetic affinities of *Eumelea*, we provisionally place the genus as *incertae sedis* (Table 2).

**Oenochrominae Guenée, 1858**

Oenochrominae has obviously been the group comprising taxa that could not easily be assigned to other subfamilies. Out of the 76 genera currently assigned to Oenochrominae, our study includes 25 genera (28 species). Three of these genera will be formally transferred to Sterrhinae (P. Sihvonen et al., 2019, unpublished data), three are here transferred to Desmobathrinae (see above, Table 2), and eight are transferred to Epidesmiinae (see below). In agreement with *Sihvonen et al. (2011)*, Oenochrominae *s.str.* grouped together in a well-supported lineage. Genera of this clade can be characterized as having robust bodies, and their male genitalia have a well-developed uncus and gnathos, broad valvae and a well-developed anellus (*Scoble & Edwards, 1990*). Common host plants are members of Proteaceae and Myrtaceae (*Holloway, 1996*). Our results strongly suggest that the genus *Oenochroma* is polyphyletic: *Oenochroma infantilis* is sister to a clade including *Dinophalus*, *Hypographa*, *Lissomma*, *Sarcinodes* and (at least) two species of *Oenochroma*. To date, 20 species have been assigned to *Oenochroma* by *Scoble (1999)*, and one additional species was described by *Hausmann et al. (2009)*, who suggested that *Oenochroma vinaria* is a species complex. We agree with *Hausmann et al. (2009)*, who pointed out the need for a major revision of *Oenochroma*.

In our phylogenetic hypothesis, *Sarcinodes* is sister to *O. orthodesma* and *O. vinaria*, the type species of *Oenochroma*. Although *Sarcinodes* and *Oenochroma* resemble each other in external morphology, a sister-group relationship between these genera has not been hypothesized before. The inclusion of *Sarcinodes* in Oenochrominae is mainly based on shared tympanal characters (*Scoble & Edwards, 1990*). However, the circular form of the lacinia, which is an apomorphy of Oenochrominae *s.str.* is missing or not apparent in *Sarcinodes* (*Holloway, 1996*). In addition, *Sarcinodes* is found in the Oriental rather than in the Australian region, where all *Oenochroma* species are distributed. A second clade of Oenochrominae *s.str.* comprises the genera *Monoctenia*, *Onycodes*, *Parepisparis*, *Antictenia*, *Arhodia*, *Gastrophora* and *Homospora*, which clustered together as the sister of *Oenochroma* and its relatives. These genera are widely recognized in sharing similar structure of the male genitalia (*Scoble & Edwards, 1990*), yet their phylogenetic relationships have never been tested. *Young (2006)* suggested the monophyly of Oenochrominae *s.str.*, however, with a poorly resolved topology and low branch support. In her study, *Parepisparis*, *Phallaria* and *Monoctenia* shared a bifid head, while in *Parepisparis* and *Onychodes*, the aedeagus was lacking caecum and cornuti. Our analysis supports these morphological similarities. *Monoctenia*, *Onycodes* and *Parepisparis* clustered together. However, a close relationship of the genera *Antictenia*, *Arhodia*, *Gastrophora* and *Homospora* has not been suggested before. Our analysis thus strongly supports the earliest definition of Oenochrominae proposed by *Guenée (1858)*, and reinforced by *Cook & Scoble (1992)*. Oenochrominae should be restricted to *Oenochroma* and related genera such as *Dinophalus*, *Hypographa*, *Lissomma*, *Sarcinodes*, *Monoctenia*, *Onycodes*, *Parepisparis*, *Antictenia*, *Arhodia*, *Gastrophora*, *Homospora*, *Phallaria* and

*Palaeodoxa*. We consider that genera included in Oenochrominae by *Scoble & Edwards (1990)*, but recovered in a lineage separate from *Oenochroma* and its close relatives in our study, belong to a hitherto unknown subfamily, which is described below.

**Epidesmiinae** Murillo-Ramos, Brehm & Sihvonen **new subfamily**
LSIDurn:lsid:zoobank.org:act:34D1E8F7-99F1-4914-8E12-0110459C2040
Type genus: *Epidesmia* Duncan & Westwood, 1841.

Material examined: Taxa included in the molecular phylogeny: *Ecphyas holopsara* Turner, 1929, *Systatica xanthastis* Lower, 1894, *Adeixis griseata* Hudson, 1903, *Dichromodes indicataria* Walker, 1866, *Phrixocomes* sp. Turner, 1930, *Abraxaphantes perampla* Swinhoe, 1890, *Epidesmia chilonaria* (Herrich-Schäffer, 1855), *Phrataria replicataria* Walker, 1866.

Most of the slender-bodied Oenochrominae, excluded from Oenochrominae *s.str.* by *Holloway (1996)*, were recovered as an independent lineage (Fig. 4) that consists of two clades: *Ec. holopsara* + *S. xanthastis* and *Ep. chilonaria* + five other genera. Branch support values from IQ-TREE strongly support the monophyly of this clade (SH-like and UFBoot2 = 100), while in RAxML the clade is moderately supported (RBS = 89). These genera have earlier been assigned to Oenochrominae *s.l.* (*Scoble & Edwards, 1990*). However, we recovered the group as a well-supported lineage independent from Oenochrominae *s.str.* and transfer them to Epidesmiinae, subfam. n. (Table 2).

Phylogenetic position: Epidesmiinae is sister to Oenochrominae *s.str.* + *Eumelea* + Geometrinae + Ennominae.

Short description of Epidesmiinae: Antennae in males unipectinate (exception: *Adeixis*), shorter towards the apex. Pectination moderate or long. Thorax and abdomen slender (unlike in Oenochrominae). Forewings with sinuous postmedial line and areole present. Forewings planiform (with wings lying flat on the substrate) in resting position, held like a triangle and cover the hindwings.

Diagnosis of Epidesmiinae: The genera included in this subfamily form a strongly supported clade with DNA sequence data from the following gene regions (exemplar *Epidesmia chilonaria* (Herrich-Schäffer, 1855)) ArgK (MK738299), Ca-ATPase (MK738690), CAD (MK738960), COI (MK739187), EF1a (MK740168), GAPDH (MK740402), MDH (MK740974) and Nex9 (MK741433). A thorough morphological investigation of the subfamily, including diagnostic characters, is under preparation.

Distribution: Most genera are distributed in the Australian region, with some species ranging into the Oriental region. *Abraxaphantes* occurs exclusively in the Oriental region.

**Geometrinae Stephens, 1829**
The monophyly of Geometrinae is strongly supported, but the number of tribes included in this subfamily is still unclear. *Sihvonen et al. (2011)* analyzed 27 species assigned to 11 tribes, followed by *Ban et al. (2018)* with 116 species in 12 tribes. *Ban et al. (2018)* synonymized nine tribes, and validated the monophyly of 12 tribes, with two new tribes Ornithospilini and Agathiini being the first two clades branching off the main lineage of

Geometrinae. Our study (168 species) validates the monophyly of 13 tribes, eleven of which were defined in previous studies: Hemitheini, Dysphaniini, Pseudoterpnini *s.str.*, Ornithospilini, Agathiini, Aracimini, Neohipparchini, Timandromorphini, Geometrini, Comibaeini, Nemoriini. One synonymization is proposed: Synchlorini Ferguson, 1969 **syn. nov.** is synonymized with Nemoriini Gumppenberg, 1887. One tribe is proposed as new: Chlorodontoperini **trib. nov.**, and one tribe (Archaeobalbini Viidalepp, 1981, **stat. rev.**) is raised from synonymy with Pseudoterpnini.

*Ban et al. (2018)* found that *Ornithospila* Warren, 1894 is sister to the rest of Geometrinae, and *Agathia* Guenée, 1858 is sister to the rest of Geometrinae minus *Ornithospila*. Although weakly supported, our results (with more species of *Agathia* sampled) placed Ornisthospilini+Agathiini together and these tribes are the sister to the rest of Geometrinae. *Chlorodontopera* is placed as an isolated lineage as shown by *Ban et al. (2018)*. Given that *Chlorodontopera* clearly forms an independent and well-supported lineage we propose the description of a new tribe Chlorodontoperini.

Chlorodontoperini Murillo-Ramos, Sihvonen & Brehm, **new tribe**
LSIDurn:lsid:zoobank.org:act:0833860E-A092-43D6-B2A1-FB57D9F7988D
Type genus: *Chlorodontopera* Warren, 1893
Material examined: Taxa in the molecular phylogeny: *Chlorodontopera discospilata* (Moore, 1867) and *Chlorodontopera mandarinata* (Leech, 1889).

Some studies (*Inoue, 1961*; *Holloway, 1996*) suggested the morphological similarities of *Chlorodontopera* Warren, 1893 with members of Aracimini. Moreover, *Holloway (1996)* considered this genus as part of Aracimini. Our results suggest a sister relationship of *Chlorodontopera* with a large clade comprising Aracimini, Neohipparchini, Timandromorphini, Geometrini, Nemoriini and Comibaenini. Considering that our analysis strongly supports *Chlorodontopera* as an independent lineage (branch support SH-like = 99 UFBoot2 = 100, RBS = 99), we introduce the monobasic tribe Chlorodontoperini. This tribe can be diagnosed by the combination of DNA data from six genetic markers (exemplar *Chlorodontopera discospilata*) CAD (MG015448), COI (MG014735), EF1a (MG015329), GAPDH (MG014862), MDH (MG014980) and RpS5 (MG015562). *Ban et al. (2018)* did not introduce a new tribe because the relationship between *Chlorodontopera* and *Euxena* Warren, 1896 was not clear in their study. This relationship was also been proposed by *Holloway (1996)* based on similar wing patterns. Further analyses are needed to clarify the affinities between *Chlorodontopera* and *Euxena*.

The tribe Chlorodontoperini is diagnosed by distinct discal spots with pale margins on the wings, which are larger on the hindwing; a dull reddish-brown patch is present between the discal spot and the costa on the hindwing, and veins M3 and CuA1 are not stalked on the hindwing (*Ban et al., 2018*). In the male genitalia, the socii are stout and setose and the lateral arms of the gnathos are developed, not joined. Sternite 3 of the male has setal patches (see *Holloway, 1996* for illustrations). Formal taxonomic changes are listed in Table 2.

Aracimini, Neohipparchini, Timandromorphini, Geometrini and Comibaenini were recovered as monophyletic groups. These results are in full agreement with

*Ban et al. (2018)*. However, the phylogenetic position of *Eucyclodes* Warren, 1894 is uncertain (unnamed G2). The monophyly of Nemoriini and Synchlorini is not supported. Instead, Synchlorini are nested within Nemoriini (support branch SH-like = 98.3, UFBoot2 = 91, RBS = 93). Our findings are in concordance with *Sihvonen et al. (2011)* and *Ban et al. (2018)*, but our analyses included a larger number of markers and a much higher number of taxa. Thus, we formally synonymize Synchlorini **syn. nov.** with Nemoriini (Table 2).

The monophyly of Pseudoterpnini *sensu Pitkin, Han & James (2007)* could not be recovered. Similar results were shown by *Ban et al. (2018)* who recovered Pseudoterpnini *s.l.* including all the genera previously studied by *Pitkin, Han & James (2007)*, forming a separate clade from *Pseudoterpna* Hübner, 1823 + *Pingasa* Moore, 1887. Our results showed African *Mictoschema* Prout, 1922 falling within Pseudoterpnini *s.str.*, and it is sister to *Pseudoterpna* and *Pingasa*. A second group of Pseudoterpnini *s.l.* was recovered as an independent lineage clearly separate from Pseudoterpnini *s.str.* (SH-like = 88.3, UFBoot2 = 64). *Ban et al. (2018)* did not introduce a new tribe due to the morphological similarities and difficulty in finding apomorphies of Pseudoterpnini *s.str.* In addition, their results were weakly supported. Considering that two independent studies have demonstrated the paraphyly of Pseudoterpnini *sensu* Pitkin et al. (2007), we see no reason for retaining the wide concept of this tribe. Instead, we propose the revival of the tribe status of Archaeobalbini.

Archaeobalbini Viidalepp, 1981, **status revised**

(original spelling: Archeobalbini, justified emendation in Hausmann (1996))

Type genus: *Archaeobalbis* Prout, 1912 (synonymized with *Herochroma* Swinhoe, 1893 in *Holloway (1996)*)

Material examined: *Herochroma curvata* Han & Xue, 2003, *H. baba* Swinhoe 1893, *Metallolophia inanularia* Han & Xue, 2004, *M. cuneataria* Han & Xue, 2004, *Actenochroma muscicoloraria* (Walker, 1862), *Absala dorcada* Swinhoe, 1893, *Metaterpna batangensis* Hang & Stüning, 2016, *M. thyatiraria* (Oberthür, 1913), *Limbatochlamys rosthorni* Rothschild, 1894, *Psilotagma pictaria* (Moore, 1888), *Dindica para* Swinhoe, 1893, *Dindicodes crocina* (Butler, 1880), *Lophophelma erionoma* (Swinhoe, 1893), *L. varicoloraria* (Moore, 1868), *L. iterans* (Prout, 1926) and *Pachyodes amplificata* (Walker, 1862).

This lineage splits into four groups: *Herochroma* Swinhoe, 1893 + *Absala* Swinhoe, 1893 + *Actenochroma* Warren, 1893 is the sister lineage of the rest of Archaeobalbini that were recovered as three clades with unresolved relationships comprising the genera *Limbatochlamys* Rothschild, 1894, *Psilotagma* Warren, 1894, *Metallolophia* Warren, 1895, *Metaterpna* Yazaki, 1992, *Dindica* Warren, 1893, *Dindicodes* Prout, 1912, *Lophophelma* Prout, 1912 and *Pachyodes* Guenée, 1858. This tribe can be diagnosed by the combination of DNA data from six genetic markers, see for instance *Pachyodes amplificata* CAD (MG015522), COI (MG014818), EF1a (MG015409), GAPDH (MG014941), MDH (MG015057) and RpS5 (MG015638). Branch support values in IQ-TREE confirm the

monophyly of this clade (SH-like = 88.3, UFBoot2 = 64). GenBank accession numbers are shown in Supplementary Material. A morphological diagnosis requires further research.

*Xenozancla* Warren, 1893 (unnamed G3) is sister to the clade comprising Dysphaniini and Pseudoterpnini *s.str*. Sihvonen et al. (2011) did not include *Xenozancla* in their analyses and suggested a sister relationship of Dysphaniini and Pseudoterpnini, but with low support. According to Ban et al. (2018), *Xenozancla* is more closely related to Pseudoterpnini *s.str*. than to Dysphaniini. However, due to low support, Ban et al. (2018) did not propose a taxonomic assignment for *Xenozancla*, which is currently not assigned to a tribe. Although our IQ-TREE results show that *Xenozancla* is sister to a clade comprising Dysphaniini and Pseudoterpnini *s.str*., the RAxML analysis did not recover the same phylogenetic relationships. Instead, Dysphaniini + Pseudoterpnini *s.str*. are found to be sister taxa, but *Xenozancla* is placed close to *Rhomborista monosticta* (Wehrli, 1924). As in Ban et al. (2018), our results do not allow us to reach a conclusion about the phylogenetic affinities of these tribes, due to low support of nodes.

The Australian genus *Crypsiphona* Meyrick, 1888 (unnamed G4) was placed close to Hemitheini. *Crypsiphona* has been assigned to Pseudoterpnini (e. g. Pitkin, Han & James, 2007, Õunap & Viidalepp, 2009), but is recovered as a separate lineage in our tree. Given the isolated position of *Crypsiphona*, the designation of a new tribe could be considered, but due to low support of nodes in our analyses, further information (including morphology) is needed to confirm the phylogenetic position of this genus. In our phylogenetic hypothesis, a large clade including the former tribes Lophochoristini, Heliotheini, Microloxiini, Thalerini, Rhomboristini, Hemistolini, Comostolini, Jodini and Thalassodini is recovered as sister to the rest of Geometrinae. These results are in full agreement with Ban et al. (2018), who synonymized all of these tribes with Hemitheini. Although the monophyly of Hemitheini is strongly supported, our findings recovered only a few monophyletic subtribes. For example, genera placed in Hemitheina were intermixed with those belonging to Microloxiina, Thalassodina and Jodina. Moreover, many genera which were unassigned to tribe, were recovered as belonging to Hemitheini. Our findings recovered *Lophostola* Prout, 1912 as sister to all Hemitheini. These results are quite different from those found by Ban et al. (2018) who suggested Rhomboristina as being sister to the rest of Hemitheini. In contrast, our results recovered Rhomboristina mingled with Hemistolina. These different results are probably influenced by the presence of African and Madagascan *Lophostola* in our analysis. In our opinion the subtribe concept, as applied in Hemitheini earlier, is not practical and we do not advocate its use in geometrid classification.

**Ennominae Duponchel, 1845**

Ennominae are the most species-rich subfamily of geometrids. The loss of vein M2 on the hindwing is probably the best apomorphy (Holloway, 1994), although vein M2 is present as tubular in a few ennomine taxa (Staude, 2001; Skou & Sihvonen, 2015). Ennominae are a morphologically highly diverse subfamily, and attempts to find further synapomorphies shared by all major tribal groups have failed.

The number of tribes as well as phylogenetic relationships among tribes are still debated (see Skou & Sihvonen, 2015 for an overview). Moreover, the taxonomic knowledge of this

subfamily in tropical regions is still poor. *Holloway (1994)* recognized 21 tribes, *Beljaev (2006)* 24 tribes, and *Forum Herbulot (2007)* 27 tribes. To date, four molecular studies have corroborated the monophyly of Ennominae (*Yamamoto & Sota, 2007*; *Wahlberg et al., 2010*; *Õunap et al., 2011*, *Sihvonen et al., 2011*), with *Young (2006)* being the only exception who found Ennominae paraphyletic. Moreover, four large-scale taxonomic revisions (without a phylogenetic hypothesis) were published by *Pitkin (2002)* for the Neotropical region, *Skou & Sihvonen (2015)*, *Müller et al. (2019)* for the Western Palaearctic region, and *Holloway (1994)* for Borneo. More detailed descriptions of taxonomic changes in Ennominae will be given by G. Brehm et al. (2019, unpublished data) and L. Murillo-Ramos et al. (2019, unpublished data). We here discuss general patterns and give details for taxonomic acts not covered in the other two papers.

Our findings recover Ennominae as a monophyletic entity, but results were not highly supported in RAxML (RBS = 67) compared to IQ-TREE (SH-Like =100, UFBoot2 = 99). The lineage comprising Geometrinae and Oenochrominae is recovered as the sister clade of Ennominae. In previous studies, *Wahlberg et al. (2010)* sampled 49 species of Ennominae, *Õunap et al. (2011)* sampled 33 species, and *Sihvonen et al. (2011)* 70 species including up to eight markers per species. All these studies supported the division of Ennominae into "boarmiine" and "ennomine" moths (*Holloway, 1994*). This grouping was proposed by *Forbes (1948)* and *Holloway (1994)*, who suggested close relationships between the tribes Boarmiini, Macariini, Cassymini and Eutoeini based on the bifid pupal cremaster and the possession of a fovea in the male forewing. The remaining tribes were defined as "ennomines" based on the loss of a setal comb on male sternum A3 and the presence of a strong furca in male genitalia. Both *Wahlberg et al. (2010)* and *Sihvonen et al. (2011)* found these two informal groupings to be reciprocally monophyletic.

In our analyses, 653 species with up to 11 markers were sampled, with an emphasis on Neotropical taxa, which so far had been poorly represented in the molecular phylogenetic analyses. Our results recovered the division into two major subclades (Fig. 6), a core set of ennomines in a well-supported clade, and a poorly supported larger clade that includes the "boarmiines" among four other lineages usually thought of as "ennomines". The traditional "ennomines" are thus not found to be monophyletic in our analyses, questioning the utility of such an informal name. Our phylogenetic hypothesis supports the validation of numerous tribes proposed previously, in addition to several unnamed clades. We validate 23 tribes (*Forum Herbulot, 2007*; *Skou & Sihvonen, 2015*): Gonodontini, Gnophini, Odontoperini, Nacophorini, Ennomini, Campaeini, Alsophilini, Wilemaniini, Prosopolophini, Diptychini, Theriini, Plutodini, Palyadini, Hypochrosini, Apeirini, Epionini, Caberini, Macariini, Cassymini, Abraxini, Eutoeini and Boarmiini. We hereby propose one new tribe: Drepanogynini **trib. nov.** (Table 2). Except for the new tribe, most of the groups recovered in this study are in concordance with previous morphological classifications (*Holloway, 1994*; *Beljaev, 2006*, *2016*; *Forum Herbulot, 2007*; *Skou & Sihvonen, 2015*; *Müller et al., 2019*).

Five known tribes and two further unnamed lineages (E1, E2 in Fig. 6) form the core Ennominae: Gonodontini, Gnophini, Odontoperini, Nacophorini and Ennomini. Several Neotropical clades that conflict with the current tribal classification of Ennominae

will be described as new tribes by G. Brehm et al. (2019, unpublished data). Gonodontini and Gnophini are recovered as sister taxa. Gonodontini was defined by *Forbes (1948)* and studied by *Holloway (1994)*, who showed synapomorphies shared by *Gonodontis* Hübner, (1823), *Xylinophylla* Warren, 1898 and *Xenimpia* Warren, 1895. Our results recovered the genus *Xylinophylla* as sister of *Xenimpia* and *Psilocladia* Warren, 1898. *Psilocladia* is an African genus currently unassigned to tribe (see *Sihvonen, Staude & Mutanen, 2015* for details). Considering the strong support and that the facies and morphology are somewhat similar to other analyzed taxa in Gonodontini, we formally include *Psilocladia* in Gonodontini (Table 2). Gnophini are monophyletic and we formally transfer the African genera *Oedicentra* Warren, 1902 and *Hypotephrina* Janse, 1932, from unassigned to Gnophini (Table 2). The total number of species, and number of included genera in Gnophini are still uncertain (*Skou & Sihvonen, 2015*; *Müller et al., 2019*). Based on morphological examination, *Beljaev (2016)* treated Angeronini as a synonym of Gnophini. The costal projection on male valva bearing a spine or group of spines was considered as a synapomorphy of the group. Using molecular data, *Yamamoto & Sota (2007)* showed a close phylogenetic relationship between *Angerona* Duponchel, 1829 (Angeronini) and *Chariaspilates* Wehrli, 1953 (Gnophini). Similar results were shown by *Sihvonen et al. (2011)* who recovered *Angerona* and *Charissa* Curtis, 1826 as sister taxa, and our results also strongly support treating Angeronini as synonym of Gnophini.

*Holloway (1994)* suggested close affinities among Nacophorini, Azelinini and Odontoperini on the basis of larval characters. In a morphology-based phylogenetic study, *Skou & Sihvonen (2015)* suggested multiple setae on the proleg on A6 of the larvae as a synapomorphy of the group. Our results also support a close relationship of Nacophorini, Azelinini and Odontoperini. These clades will be treated in more detail by G. Brehm et al. (2019, unpublished data).

Following the ideas of *Pitkin (2002)*, *Beljaev (2008)* synonymized the tribes Ourapterygini and Nephodiini with Ennomini. He considered the divided vinculum in male genitalia and the attachment of muscles *m*3 as apomorphies of the Ennomini, but did not provide a phylogenetic analysis. *Sihvonen et al. (2011)* supported Beljaev's assumptions and recovered *Ennomos* Treitschke, 1825 (Ennomini), *Ourapteryx* Leach, 1814 (Ourapterygini) and *Nephodia* Hübner, 1823 (Nephodiini) as belonging to the same clade. Our comprehensive analysis confirms those previous findings and we agree with Ennomini as the valid tribal name for this large clade. This clade will be treated in more detail by G. Brehm et al. (2019, unpublished data).

Campaeini, Alsophilini, Wilemaniini and Prosopolophini grouped together in a well-supported clade (SH-like = 100, UFBoot2 = 99). Previous molecular analyses have shown an association of Colotoini [= Prosopolophini] and Wilemaniini (*Yamamoto & Sota, 2007*; *Sihvonen et al., 2011*), although no synapomorphies are known to support synonymization (*Skou & Sihvonen, 2015*). The Palaearctic genera *Compsoptera* Blanchard, 1845, *Apochima* Agassiz, 1847, *Dasycorsa* Prout, 1915, *Chondrosoma* Anker, 1854 and *Dorsispina* Nupponen & Sihvonen, 2013, are potentially part of the same complex (*Skou & Sihvonen, 2015*, Sihvonen pers. obs.), but they were not included in the current study. Campaeini is a small group including four genera with Oriental, Palaearctic and Nearctic

distribution, apparently closely related to Alsophilini and Prosopolophini, but currently accepted as a tribe (*Forum Herbulot, 2007*; *Skou & Sihvonen, 2015*). Our results support the close phylogenetic affinities among these tribes, but due to the limited number of sampled taxa, we do not propose any formal changes.

The genus *Declana* Walker, 1858 is recovered as an isolated clade sister to Diptychini. This genus is endemic to New Zealand, but to date has not been assigned to tribe. According to our results, *Declana* could well be defined as its own tribe. However, the delimitation of this tribe is beyond the scope of our paper and more genera from Australia and New Zealand should first be examined. A close relationship between Nacophorini and Lithinini was suggested by *Pitkin (2002)*, based on the similar pair of processes of the anellus in the male genitalia. Pitkin also noted a morphological similarity in the male genitalia (processes of the juxta) shared by Nacophorini and Diptychini. In a study of the Australasian fauna, *Young (2008)* suggested the synonymization of Nacophorini and Lithinini. This was further corroborated by *Sihvonen, Staude & Mutanen (2015)* who found that Diptychini were nested within some Nacophorini and Lithinini. However, none of the studies proposed formal taxonomic changes because of limited taxon sampling. In contrast, samples in our analyses cover all biogeographic regions and the results suggest that true Nacophorini is a clade which comprises almost exclusively New World species. This clade is clearly separate from Old World "nacophorines" (cf. *Young, 2003*) that are intermixed with Lithinini and Diptychini. We here formally transfer Old World nacophorines to Diptychini and synonymize Lithinini **syn. nov.** with Diptychini (Table 2). Further formal taxonomic changes in the Nacophorini complex are provided by G. Brehm et al. (2019, unpublished data).

*Theria* Hübner 1825, the only representative of Theriini in this study, clustered together with *Lomographa* Hübner, 1825 (Baptini in *Skou & Sihvonen, 2015*), in a well-supported clade, agreeing with the molecular results of *Sihvonen et al. (2011)*. The placement of *Lomographa* in Caberini (*Rindge, 1979*; *Pitkin, 2002*) is not supported by our study nor by that of *Sihvonen et al. (2011)*. The monophyly of *Lomographa* has not been tested before, but we show that one Neotropical and one Palaearctic *Lomographa* species indeed group together. Our results show that Caberini are not closely related to the Theriini + Baptini clade, unlike in earlier morphology-based hypotheses (*Rindge, 1979*; *Pitkin, 2002*). Morphologically, Theriini and Baptini are dissimilar, therefore we recognize them as valid tribes (see description and illustrations in *Skou & Sihvonen, 2015*).

According to our results, 11 molecular markers were not enough to infer phylogenetic affinities of Plutodini (represented by one species of *Plutodes*). Similar results were found by *Sihvonen et al. (2011)*, who in some analyses recovered *Plutodes* as sister of *Eumelea*. Our analyses are congruent with those findings. IQ-TREE results suggest that *Plutodes* is sister to Palyadini, but RAxML analyses recovered *Eumelea* as the most probable sister of *Plutodes*. Given that our analyses are not in agreement on the sister-group affinities of *Plutodes*, we do not make any assumptions about its phylogenetic position. Instead, we emphasize that further work needs to be done to clarify the phylogenetic positions of *Plutodes* and related groups.

Hypochrosini is only recovered in a well-defined lineage if the genera *Apeira* Gistl, 1848 (Apeirini), *Epione* Duponchel, 1829 (Epionini), *Sericosema* (Caberini), *Ithysia* (Theriini),

*Capasa* Walker, 1866 (unassigned) and *Omizodes* Warren, 1894 (unassigned) were transferred to Hypochrosini. *Skou & Sihvonen (2015)* already suggested a close association of Epionini, Apeirini and Hypochrosini. We think that synonymizing these tribes is desirable. However, due to the limited number of sampled taxa we do not propose any formal changes until more data becomes available. We do suggest, however, formal taxonomic changes for the genera *Capasa* and *Omizodes* from unassigned to Hypochrosini (Table 2).

The southern African genus *Drepanogynis* is paraphyletic and has earlier been classified as belonging in Ennomini, and later in Nacophorini (*Krüger, 2002*). In our phylogeny, it is intermixed with the genera *Sphingomima* Warren, 1899, and *Thenopa* Walker, 1855. *Hebdomophruda errans* Prout, 1917 also clusters together with these taxa, apart from other *Hebdomophruda* Warren, 1897 species, which suggests that this genus is polyphyletic. These genera form a clade sister to the lineage that comprises several Hypochrosini species. Considering that our analysis strongly supports this clade, we place *Thenopa*, *Sphingomima* and *Drepanogynis* in a tribe of their own.

Drepanogynini Murillo-Ramos, Sihvonen & Brehm **new tribe**

LSIDurn:lsid:zoobank.org:act:AA384988-009F-4175-B98C-6209C8868B93

Type genus: *Drepanogynis* Guenée, (1858)

The African genera *Thenopa*, *Sphingomima* and *Drepanogynis* appear as a strongly supported lineage (SH-like, UFBoot2 and RBS = 100). *Krüger (1997*, p. 259) proposed "Boarmiini and related tribes as the most likely sister group" for *Drepanogynis*, whereas more recently *Drepanogynis* was classified in the putative southern hemisphere Nacophorini (*Krüger, 2014*; *Sihvonen, Staude & Mutanen, 2015*). In the current phylogeny, *Drepanogynis* is isolated from Nacophorini *sensu stricto* and from other southern African genera that have earlier been considered to be closely related to it (*Krüger, 2014* and references therein). The other southern African genera appeared to belong to Diptychini in our study. The systematic position of *Drepanogynis tripartita* (Warren, 1898) has earlier been analyzed in a molecular study (*Sihvonen, Staude & Mutanen, 2015*). The taxon grouped together with the Palaearctic species of the tribes Apeirini, Theriini, Epionini and putative Hypochrosini. *Sihvonen, Staude & Mutanen (2015)* noted that *Argyrophora trofonia* (Cramer, 1779) (representing *Drepanogynis* group III *sensu Krüger, 1999*) and *Drepanogynis tripartita* (representing *Drepanogynis* group IV *sensu Krüger, 2002*) did not group together, but no formal changes were proposed. Considering that the current analysis strongly supports the placement of *Drepanogynis* and related genera in an independent lineage, and the aforementioned taxa in the sister lineage (Apeirini, Theriini, Epionini and putative Hypochrosini) have been validated at tribe-level, we place *Drepanogynis* and related genera in a tribe of their own.

Material examined and taxa included: *Drepanogynis mixtaria* (Guenée, 1858), *D. tripartita*, *D. determinata* (Walker, 1860), *D. arcuifera* Prout, 1934, *D. arcuatilinea Krüger, 2002*, *D. cnephaeogramma* (Prout, 1938), *D. villaria* (Felder & Rogenhofer, 1875), "*Sphingomima*" discolucida Herbulot, 1995 (genus combination uncertain, see taxonomic

notes below), *Thenopa diversa* Walker, 1855, *"Hebdomophruda" errans* Prout, 1917 (genus combination uncertain, see taxonomic notes below).

Taxonomic notes: We choose *Drepanogynis* Guenée, 1858 as the type genus for Drepanogynini, although it is not the oldest valid name (ICZN Article 64), because extensive literature has been published on *Drepanogynis* (*Krüger, 1997*, *1998*, *1999*, *2014*), but virtually nothing exists on *Thenopa*, Walker, 1855, except the original descriptions of its constituent species. Current results show the urgent need for more extensive phylogenetic studies within Drepanogynini. *Thenopa* and *Sphingomima* are embedded within *Drepanogynis*, rendering it paraphyletic, but our taxon coverage is too limited to propose formal changes in this species-rich group. Drepanogynini, as defined here, are distributed in sub-Saharan Africa. *Drepanogynis sensu Krüger (1997*, *1998*, *1999*, *2014*) includes over 150 species and it ranges from southern Africa to Ethiopia (*Krüger, 2002*, *Vári, Kroon & Krüger, 2002*), whereas the genera *Sphingomima* (10 species) and *Thenopa* (four species) occur in Central and West Africa (*Scoble, 1999*). *Sphingomima* and *Thenopa* are externally similar, so the recovered sister-group relationship in the current phylogeny analysis was anticipated. In the current analysis, *Hebdomophruda errans* Prout, 1917 is isolated from other analyzed *Hebdomophruda* species (the others are included in Diptychini), highlighting the need for additional research. *Krüger (1997*, *1998*) classified the genus *Hebdomophruda* into seven species groups on the basis of morphological characters, and *H. errans* group is one of them (*Krüger, 1998*). We do not describe a new genus for the taxon *errans*, nor do we combine it with any genus in the Drepanogynini, highlighting its uncertain taxonomic position (*incertae sedis*) pending more research. In the current analysis, *Sphingomima discolucida* Herbulot, 1995 is transferred from unassigned tribus combination to Drepanogynini, but as the type species of *Sphingomima* (*S. heterodoxa* Warren, 1899) was not analyzed, we do not transfer the entire genus *Sphingomima* into Drepanogynini. We highlight the uncertain taxonomic position of the taxon *discolucida*, acknowledging that it may eventually be included again in *Sphingomima* if the entire genus should be transferred to Drepanogynini.

Diagnosis: Drepanogynini can be diagnosed by the combination of DNA data with up to 11 genetic markers (exemplar *Drepanogynis mixtaria* (Guenée, 1858)) ArgK (MK738841), COI (MK739615), EF1a (MK739960), IDH (MK740862), MDH (MK741181), Nex9 (MK741630), RpS5 (MK741991) and Wingless (MK742540). In the light of our phylogenetic results, the *Drepanogynis* group of genera, as classified earlier (*Krüger, 2014*), is split between two unrelated tribes (Drepanogynini and Diptychini). More research is needed to understand how other *Drepanogynis* species and the *Drepanogynis* group of genera *sensu Krüger (1997*, *1998*, *1999*, *2014)* (at least 11 genera), should be classified.

Boarmiini are the sister group to a clade that comprises Macariini, Cassymini, Abraxini and Eutoeini. We found that many species currently classified as Boarmiini are scattered throughout Ennominae. Boarmiini *s.str.* are strongly supported but are technically not monophyletic because of a large number of genera which need to be formally transferred from other tribes to Boarmiini (G. Brehm et al., 2019, unpublished data for Neotropical taxa and L. Murillo-Ramos et al., 2019, unpublished data for other taxa). The results are

principally in concordance with *Jiang et al. (2017)*, who supported the monophyly of Boarmiini but with a smaller number of taxa.

The divided valva in male genitalia was suggested as a synapomorphy of Macariini + Cassymini + Eutoeini by *Holloway (1994)*. In addition, he proposed the inclusion of Abraxini in Cassymini. Although our findings support a close relationship, this group requires more study and a more extensive sampling effort. Similar findings were provided by *Jiang et al. (2017)* who suggested more extensive sampling to study the evolutionary relationships of these tribes.

### Orthostixinae Meyrick, 1892

Orthostixinae were not included in our study. *Sihvonen et al. (2011)* showed this subfamily as deeply embedded within Ennominae, but unfortunately it was not represented by the type genus of the subfamily. These results agree with *Holloway (1996)* who examined *Orthostixis* Hübner, (1823) and suggested the inclusion in Ennominae despite the full development of hindwing vein M2, the presence of a forewing areole and the very broad base of the tympanal ansa. We sampled the species *Naxa textilis* (Walker, 1856) and *Orthostixis cribraria* (Hübner, 1799), but only three and one marker were successfully sequenced for these samples, respectively. We included these species in the preliminary analyses but results were so unstable that we excluded them from the final analysis. Further research including fresh material and more genetic markers are needed to investigate the position of Orthostixinae conclusively.

## CONCLUSIONS

This study elucidated important evolutionary relationships among major groups within Geometridae. The monophyly of the subfamilies and the most widely accepted tribes were tested. We found strong support for the traditional concepts of Larentiinae, Geometrinae and Ennominae. Sterrhinae also becomes monophyletic when *Ergavia*, *Ametris* and *Macrotes*, currently placed in Oenochrominae, are formally transferred to Sterrhinae. The concepts of Oenochrominae and Desmobathrinae required major revision and, after appropriate rearrangements, these groups also form monophyletic subfamily-level entities. Archiearinae are monophyletic with the transfer of *Dirce* and *Acalyphes* to Ennominae. We treat Epidesmiinae as a new subfamily.

This study proposes the recognition of eight monophyletic geometrid subfamilies. Many geometrid tribes were recovered para- or polyphyletic. We attempted to address the needed taxonomic changes, in order to favor taxonomic stability of the subfamilies and many tribes, even if in an interim way, to allow other researchers to use an updated higher-taxonomic structure that better reflects our current understanding of geometrid phylogeny. Although we included a large number of new taxa, in our study, many clades remain poorly represented. This is particularly true for taxa from tropical Africa and Asia. Tribes in special need of reassessment include Eumeleini, Plutodini, Eutoeini, Cassymini and Abraxini. We hope the phylogenetic hypotheses shared here will open new paths of inquiry across Geometridae. Morphological synapomorphies have not yet been identified for many of the re- and newly defined higher taxa circumscribed by our 11-gene

data set. Likewise, there is great need, across the family, to begin the work of mapping behavioral and life history attributes to the clades identified in this work.

## ACKNOWLEDGEMENTS

Harri Sihvonen (Finland) is thanked for preparing extensive African materials for the study. We are grateful to Cathy Byrne (Hobart, Australia), B.C. Schmidt (Canada, Ottawa), Alfred Moser (Curitiba, Brazil), Rolf Mörtter (Karlsruhe, Germany), Daniel Bolt (Domat/Ems, Switzerland), Florian Bodner (Vienna, Austria), Dominik Rabl (Vienna, Austria), Aare Lindt (Tallinn, Estonia), Luis Parra (Concepción, Chile), Andreas Kopp (St. Margarethen, Switzerland), Stefan Naumann (Berlin, Germany), Jaan Viidalepp (Tartu, Estonia) for providing samples for this study. We thank John Chainey, Geoff Martin and Linda Pitkin at the NHM (London) for providing access to the collections and photographs of Neotropical Ennominae moths. We thank David Wagner, Andreas Zwick and Kevin Keegan for constructive comments on the manuscript.

### Funding

Niklas Wahlberg received funding from the Academy of Finland (Grant No. 265511) and the Swedish Research Council (Grant No. 2015-04441). Leidys Murillo-Ramos received funding from Colciencias, 756-2016 and Universidad de Sucre, Colombia. Hamid Reza Ghanavi was funded from the European Union's Horizon 2020 research and innovation program under the Marie Sklodowska-Curie grant agreement no. 642241 (BIG4). Sille Holm, Erki Õunap, Andro Truuverk and Toomas Tammaru were supported by institutional research funding IUT (IUT20-33) of the Estonian Ministry of Education and Research. Gunnar Brehm received funding for fieldwork in Peru (DFG grant Br 2280/6-1) and for visits to the NHM (SYNTHESYS grant GB TAF1048 and 6817). The funders had no role in study design, data collection and analysis, decision to publish, or preparation of the manuscript.

### Grant Disclosures

The following grant information was disclosed by the authors:
Academy of Finland: 265511.
Swedish Research Council: 2015-04441.
Colciencias: 756-2016.
Universidad de Sucre, Colombia.
European Union's Horizon 2020 research and innovation program under the Marie Sklodowska-Curie grant agreement no. 642241 (BIG4).
Estonian Ministry of Education and Research: IUT20-33.
Funding for fieldwork in Peru: DFG grant Br 2280/6-1) and for visits to the NHM: SYNTHESYS grant GB TAF1048 and 6817.

## Competing Interests

All the authors declare that they have no competing interests.

## Author Contributions

- Leidys Murillo-Ramos conceived and designed the experiments, performed the experiments, analyzed the data, contributed reagents/materials/analysis tools, prepared figures and/or tables, authored or reviewed drafts of the paper, approved the final draft.
- Gunnar Brehm conceived and designed the experiments, analyzed the data, contributed reagents/materials/analysis tools, prepared figures and/or tables, authored or reviewed drafts of the paper, approved the final draft.
- Pasi Sihvonen conceived and designed the experiments, analyzed the data, contributed reagents/materials/analysis tools, prepared figures and/or tables, authored or reviewed drafts of the paper, approved the final draft.
- Axel Hausmann contributed reagents/materials/analysis tools, authored or reviewed drafts of the paper, approved the final draft.
- Sille Holm performed the experiments, authored or reviewed drafts of the paper, approved the final draft.
- Hamid Reza Ghanavi performed the experiments, analyzed the data, contributed reagents/materials/analysis tools, authored or reviewed drafts of the paper, approved the final draft.
- Erki Õunap performed the experiments, contributed reagents/materials/analysis tools, authored or reviewed drafts of the paper, approved the final draft.
- Andro Truuverk performed the experiments, authored or reviewed drafts of the paper, approved the final draft.
- Hermann Staude contributed reagents/materials/analysis tools, authored or reviewed drafts of the paper, approved the final draft.
- Egbert Friedrich contributed reagents/materials/analysis tools, authored or reviewed drafts of the paper, approved the final draft.
- Toomas Tammaru contributed reagents/materials/analysis tools, authored or reviewed drafts of the paper, approved the final draft.
- Niklas Wahlberg conceived and designed the experiments, analyzed the data, contributed reagents/materials/analysis tools, prepared figures and/or tables, authored or reviewed drafts of the paper, approved the final draft.

## DNA Deposition

The following information was supplied regarding the deposition of DNA sequences:

The sequences described here are accessible via GenBank with the following accession numbers:

MK738162–MK738576; MK738577–MK738903; MK738904–MK739052; MK739053–MK739692; MK739693–MK740306; MK740307–MK740765; MK740766–MK740930; MK740931–MK741338; MK741339–MK741692; MK741693–MK742127; MK742128–MK742716.

## Data Availability

The raw analyses are available in the Supplemental Files. The RAxML tree was used to compare some clades recovered in IQTREE.

## New Species Registration

The following information was supplied regarding the registration of a newly described species:

Publication LSID: urn:lsid:zoobank.org:pub:662A9A18-B620-45AA-B4B1-326086853316

Epidesmiinae LSID: urn:lsid:zoobank.org:act:34D1E8F7-99F1-4914-8E12-0110459C2040

Tribe Chlorodontoperini LSID: urn:lsid:zoobank.org:act:0833860E-A092-43D6-B2A1-FB57D9F7988D

Tribe Drepanogynini LSID: urn:lsid:zoobank.org:act:AA384988-009F-4175-B98C-6209C8868B93.

## Supplemental Information

Supplemental information for this article can be found online at http://dx.doi.org/10.7717/peerj.7386#supplemental-information.

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

Nomenclature. Fourth Edition. *Available at https://www.iczn.org/the-code/the-international-
code-of-zoological-nomenclature/the-code-online/*.

**Jiang N, Li X, Hausmann A, Cheng R, Xue DY, Han HX. 2017.** A molecular phylogeny of the
Palaearctic and Oriental members of the tribe Boarmiini (Lepidoptera: Geometridae:
Ennominae). *Invertebrate Systematics* **31(4)**:427–441 DOI 10.1071/IS17005.

**Kalyaanamoorthy S, Minh BQ, Wong TKF, von Haeseler A, Jermin LS. 2017.** ModelFinder: fast
model selection for accurate phylogenetic estimates. *Nature Methods* **14(6)**:587–589
DOI 10.1038/nmeth.4285.

**Krüger M. 1997.** Revision of Afrotropical Ennominae of the *Drepanogynis* group I: the genus *Hebdomophruda* Warren, Part 1. *Annals of the Transvaal Museum* **36**:257–291.

**Krüger M. 1998.** Revision of Afrotropical Ennominae of the *Drepanogynis* group II: the genus *Hebdomophruda* Warren, Part 2. *Annals of the Transvaal Museum* **36**:333–349.

**Krüger M. 1999.** Revision of Afrotropical Ennominae of the *Drepanogynis* group III: the genera *Argyrophora* Guenée, *Pseudomaenas* Prout and *Microligia* Warren. *Annals of the Transvaal Museum* **36**:427–496.

**Krüger M. 2002.** Revision of Afrotropical Ennominae of the *Drepanogynis* group IV: the genus Drepanogynis Guenée (Lepidoptera: Geometridae). *Transvaal Museum Monograph* **13**:1–220 incl. 442 figs.

**Krüger M. 2014.** A revision of the *Mauna* Walker, 1865 and *Illa* Warren, 1914 group of genera (Lepidoptera: Geometridae: Ennominae: Nacophorini). *Annals of the Ditsong National Museum of Natural History* **4**:77–173.

**Lanfear R, Calcott B, Ho SYW, Guindon S. 2012.** PartitionFinder: combined selection of partitioning schemes and substitution models for phylogenetic analyses. *Molecular Biology and Evolution* **29(6)**:1695–1701 DOI 10.1093/molbev/mss020.

**McQuillan PB, Edwards ED. 1996.** Geometroidea. In: Nielsen ES, Edwards TE, Rangsi TV, eds. *Checklist of the Lepidoptera of Australia*. Clayton: CSIRO Publishing.

**Meyrick E. 1889.** Revision of Australian Lepidoptera. *Proceedings of the Linnean Society of New South Wales* **41**:1117–1216 DOI 10.5962/bhl.part.15082.

**Miller MA, Pfeiffer W, Schwartz T. 2010.** Creating the CIPRES science gateway for inference of large phylogenetic trees. In: *Proceedings of the Gateway Computing Environments Workshop (GCE)*, New Orleans, LA, 1–8. *Available at* http://www.phylo.org (accessed June 2018).

**Minet J, Scoble MJ. 1999.** The Drepanoid/Geometroid assemblage. In: Kristensen NP, ed. *Handbook of Zoology, part 35, Lepidoptera, Moths and Butterflies, Vol. 1, Evolution, Systematics, and Biogeography*. Berlin: De Gruyter, 301–320.

**Mironov V. 2003.** Larentiinae II (Perizomini and Eupitheciini). In: Hausmann A, ed. *The Geometrid Moths of Europe* 4. Stenstrup: Apollo Books, 1–463.

**Müller B, Erlacher S, Hausmann A, Rajaei H, Sihvonen P, Skou P. 2019.** Ennominae II. In: Hausmann A, Sihvonen P, Rajaei H, Skou P, eds. *Geometrid Moths of Europe*. Vol. 6. Leiden: Brill, 906.

**Nguyen L-T, Schmidt HA, von Haeseler A, Minh BQ. 2015.** IQ-TREE: a fast and effective stochastic algorithm for estimating maximum likelihood phylogenies. *Molecular Biology and Evolution* **32(1)**:268–274 DOI 10.1093/molbev/msu300.

**Õunap E, Javoiš J, Viidalepp J, Tammaru T. 2011.** Phylogenetic relationships of selected European Ennominae (Lepidoptera: Geometridae). *European Journal of Entomology* **108(2)**:267–273 DOI 10.14411/eje.2011.036.

**Õunap E, Viidalepp J. 2009.** Description of *Crypsiphona tasmanica* sp. nov. (Lepidoptera: Geometridae: Geometrinae), with notes on limitations in using DNA barcodes for delimiting species. *Australian Journal of Entomology* **48(2)**:113–124 DOI 10.1111/j.1440-6055.2009.00695.x.

**Õunap E, Viidalepp J, Saarma U. 2008.** Systematic position of Lythriini revised: transferred from Larentiinae to Sterrhinae (Lepidoptera, Geometridae). *Zoologica Scripta* **37(4)**:405–413 DOI 10.1111/j.1463-6409.2008.00327.x.

**Õunap E, Viidalepp J, Truuverk A. 2016.** Phylogeny of the subfamily Larentiinae (Lepidoptera: Geometridae): integrating molecular data and traditional classifications. *Systematic Entomology* **21(4)**:824–843 DOI 10.1111/syen.12195.

Peña C, Malm T. 2012. VoSeq: a voucher and DNA sequence web application. *PLOS ONE* 7(6): e39071 DOI 10.1371/journal.pone.0039071.

Pitkin LM. 1996. Neotropical emerald moths: a review of the genera (Lepidoptera: Geometridae, Geometrinae). *Zoological Journal of the Linnean Society* 118(4):309–440 DOI 10.1111/j.1096-3642.1996.tb01268.x.

Pitkin L. 2002. Neotropical Ennomine moths: a review of the genera (Lepidoptera: Geometridae). *Zoological Journal of the Linnean Society* 135(2–3):121–401 DOI 10.1046/j.1096-3642.2002.01200.x.

Pitkin LM, Han HX, James S. 2007. Moths of the tribe Pseudoterpnini (Geometridae: Geometrinae): a review of the genera. *Zoological Journal of the Linnean Society* 150(2):343–412 DOI 10.1111/j.1096-3642.2007.00287.x.

Pitkin B, Jenkins P. 2004. Butterflies and moths of the world, generic names and their type-species. *Available at* http://www.nhm.ac.uk/our-science/data/butmoth/ (accessed 29 August 2018).

Rajaei H, Greve C, Letsch H, Stüning D, Wahlberg N, Minet J, Misof B. 2015. Advances in Geometroidea phylogeny, with characterization of a new family based on *Pseudobiston pinratanai* (Lepidoptera, Glossata). *Zoologica Scripta* 44(4):418–436 DOI 10.1111/zsc.12108.

Rambaut A. 2012. Figtree 1.4.0. *Available at* http://tree.bio.ed.ac.uk/software/figtree/ (accessed 3 August 2018).

Ratnasingham S, Hebert PDN. 2007. BOLD: the barcode of life data systems. *Molecular Ecology Notes* 7(3):355–364 DOI 10.1111/j.1471-8286.2007.01678.x.

Regier JC, Mitter C, Zwick A, Bazinet AL, Cummings MP, Kawahara AY, Sohn J-C, Zwickl DJ, Cho S, Davis DR, Baixeras J, Brown J, Parr C, Weller S, Lees DC, Mitter KT. 2013. A large-scale, higher-level, molecular phylogenetic study of the insect order Lepidoptera (moths and butterflies). *PLOS ONE* 8(3):e58568 DOI 10.1371/journal.pone.0058568.

Regier JC, Zwick A, Cummings MP, Kawahara AY, Cho S, Weller S, Roe A, Baixeras J, Brown JW, Parr C, Davis DR, Epstein M, Hallwachs W, Hausmann A, Janzen DH, Kitching IJ, Solis MA, Yen SH, Bazinet AL, Mitter C. 2009. Toward reconstructing the evolution of advanced moths and butterflies (Lepidoptera: Ditrysia): an initial molecular study. *BMC Evolutionary Biology* 9(1):280 DOI 10.1186/1471-2148-9-280.

Rindge FH. 1979. A revision of the North American moths of the genus *Lomographa* (Lepidoptera, Geometridae). *American Museum Novitates* 2673:1–18.

Rota J. 2011. Data partitioning in Bayesian analysis: molecular phylogenetics of metalmark moths (Lepidoptera: Choreutidae). *Systematic Entomology* 36(2):317–329 DOI 10.1111/j.1365-3113.2010.00563.x.

Scoble MJ. 1992. *Lepidoptera: form function and diversity*. Oxford: Oxford University Press.

Scoble MJ. 1999. *Geometrid Moths of theWorld: a catalogue (Lepidoptera, Geometridae)* 1, 2. Collingwood: CSIRO.

Scoble MJ, Edwards ED. 1990. *Parepisparis* Bethune-Baker and the composition of the Oenochrominae (Lepidoptera: Geometridae). *Entomologica Scandinavica* 20(4):371–399 DOI 10.1163/187631289X00375.

Scoble MJ, Hausmann A. 2007. *Online list of valid and available names of the Geometridae of the world. Available at* http://www.lepbarcoding.org/geometridae/species_checklists.php.

Sihvonen P, Kaila L. 2004. Phylogeny and tribal classification of Sterrhinae with emphasis on delimiting Scopulini (Lepidoptera: Geometridae). *Systematic Entomology* 29(3):324–358 DOI 10.1111/j.0307-6970.2004.00248.x.

**Sihvonen P, Mutanen M, Kaila L, Brehm G, Hausmann A, Staude HS. 2011.** Comprehensive molecular sampling yields a robust phylogeny for geometrid moths (Lepidoptera: Geometridae). *PLOS ONE* **6(6)**:e20356 DOI 10.1371/journal.pone.0020356.

**Sihvonen P, Staude H. 2011.** Geometrid moth *Afrophyla vethi* (Snellen, 1886) transferred from Oenochrominae to Sterrhinae (Lepidoptera: Geometridae). *Metamorphosis* **22**:102–113.

**Sihvonen P, Staude HS, Mutanen M. 2015.** Systematic position of the enigmatic African cycad moths: an integrative approach to a nearly century old problem (Lepidoptera: Geometridae, Diptychini). *Systematic Entomology* **40(3)**:606–627 DOI 10.1111/syen.12125.

**Skou P, Sihvonen P. 2015.** *The Geometrid moths of Europe. Vol. 5: Ennominae I.* Stenstrup: Apollo Books.

**Stamatakis A. 2014.** RAxML version 8: a tool for phylogenetic analysis and post-analysis of large phylogenies. *Bioinformatics* **30(9)**:1312–1313 DOI 10.1093/bioinformatics/btu033.

**Stamatakis A, Hoover P, Rougemont J. 2008.** A rapid bootstrap algorithm for the RAxML Web servers. *Systematic Biology* **57(5)**:758–771 DOI 10.1080/10635150802429642.

**Staude HS. 2001.** A revision of the genus *Callioratis* Felder (Lepidoptera: Geometridae: Diptychinae). *Metamorphosis* **12**:125–156.

**Staude H, Sihvonen P. 2014.** Revision of the African geometrid genus *Zerenopsis* C. &. R. Felder-moths with peculiar life histories and mating behaviors (Geometridae: Ennominae: Diptychini). *Metamorphosis* **25**:11–55.

**Strutzenberger P, Brehm G, Gottsberger B, Bodner F, Seifert CL, Fiedler K. 2017.** Diversification rates, host plant shifts and an updated molecular phylogeny of Andean *Eois* moths (Lepidoptera: Geometridae). *PLOS ONE* **12(12)**:e018843 DOI 10.1371/journal.pone.0188430.

**Trifinopoulos J, Minh B. 2018.** IQ-TREE manual: frequently asked questions. *Available at http://www.iqtree.org/doc/Frequently-Asked-Questions* (accessed 13 August 2018).

**Van Nieukerken EJ, Kaila L, Kitching IJ, Kristensen NP, Lees DC, Minet J, Mitter C, Mutanen M, Regier JC, Simonsen TJ, Wahlberg N, Yen S, Zahiri R, Adamski D, Baixeras J, Bartsch D, Bengtsson BÅ, Brown JW, Bucheli SR, Davis DR, Prins J de, De Prins W, Epstein ME, Gentili-Poole P, Gielis C, Hättenschwiler P, Hausmann A, Holloway JD, Kallies A, Karsholt O, Kawahara AY, Koster J, Kozlov M, Lafontaine JD, Lamas G, Landry J, Lee S, Nuss M, Park K, Penz C, Rota J, Schintlmeister A, Schmidt BC, Sohn J, Solis MA, Tarmann GM, Warren AD, Weller S, Yakovlev RV, Zolotuhin VV, Zwick A. 2011.** Order Lepidoptera Linnaeus, 1758. In: Zhang, Z.-Q. (Ed.), Animal biodiversity: an outline of higher-level classification and survey of taxonomic richness. *Zootaxa* **3148**:212–221.

**Viidalepp J. 2006.** Cladistic analysis of the subfamily Larentiinae. In: Hausmann A, McQuillan P, eds. Proceedings of the Forum Herbulot 2006. Integration of molecular, ecological and morphological data: recent progress towards the higher classification of the Geometridae (Hobart, 19–20 January 2006). *Spixiana* **29**:202–203.

**Viidalepp J. 2011.** A morphological review of tribes in Larentiinae (Lepidoptera: Geometridae). *Zootaxa* **3136(1)**:1–44 DOI 10.11646/zootaxa.3136.1.1.

**Vári L, Kroon DM, Krüger M. 2002.** *Classification and checklist of the species of Lepidoptera recorded in Southern Africa.* Chatswood: Simple Solutions.

**Wahlberg N, Peña C, Ahola M, Wheat CW, Rota J. 2016.** PCR primers for 30 novel gene regions in the nuclear genomes of Lepidoptera. *ZooKeys* **596**:129–141 DOI 10.3897/zookeys.596.8399.

**Wahlberg N, Snäll N, Viidalepp J, Ruohomäki K, Tammaru T. 2010.** The evolution of female flightlessness among Ennominae of the Holarctic forest zone (Lepidoptera, Geometridae). *Molecular Phylogenetics and Evolution* **55(3)**:929–938 DOI 10.1016/j.ympev.2010.01.025.

**Wahlberg N, Wheat CW. 2008.** Genomic outposts serve the phylogenomic pioneers: designing novel nuclear markers for genomic DNA extractions of Lepidoptera. *Systematic Biology* **57(2)**:231–242 DOI 10.1080/10635150802033006.

**Yamamoto S, Sota T. 2007.** Phylogeny of the Geometridae and the evolution of winter moths inferred from a simultaneous analysis of mitochondrial and nuclear genes. *Molecular Phylogenetics and Evolution* **44(2)**:711–723 DOI 10.1016/j.ympev.2006.12.027.

**Young CJ. 2003.** The place of the Australian Nacophorini in the Geometridae. *Spixiana* **26**:199–200.

**Young CJ. 2006.** Molecular relationships of the Australian Ennominae (Lepidoptera: Geometridae) and implications for the phylogeny of the Geometridae from molecular and morphological data. *Zootaxa* **1264(1)**:1–147 DOI 10.11646/zootaxa.1264.1.1.

**Young CJ. 2008.** Characterization of the Australian Nacophorini using adult morphology, and phylogeny of the Geometridae based on morphological characters. *Zootaxa* **1736(1)**:1–141 DOI 10.11646/zootaxa.1736.1.1.