# Peer review of "A comprehensive molecular phylogeny of Geometridae (Lepidoptera) with a focus on enigmatic small subfamilies"

_PeerJ, doi:10.7717/peerj.7386_

## Round 0.1 · original submission · Minor Revisions

Dear Dr. Murillo Ramos and colleagues:

Thanks for submitting your manuscript to PeerJ. I have now received two independent reviews of your work, and as you will see, both are very favorable. Well done! Nonetheless, the reviewers raised some relatively minor concerns about the research, and areas where the manuscript can be improved. I agree with the reviewers, and thus feel that their concerns should be adequately addressed before moving forward.

Aside from the criticisms raised by the reviewers in their reports, be sure to thoroughly evaluate the marked-up manuscripts kindly provided by both reviewers.

Therefore, I am recommending that you revise your manuscript accordingly, taking into account all of the issues raised by the reviewers. I do believe that your manuscript will be ready for publication once these issues are addressed.

Good luck with your revision,

-joe

·

Basic reporting

No comments. See my full review.

Experimental design

See full review.

Validity of the findings

See full review

Additional comments

I read the manuscript 1.5x: my attention flagged in the taxonomically arcane Discussion on the second read. The science is sound. It represents an enormous amount of work and is abundantly worthy of publication. Foremost, the data set is among the largest and most comprehensive molecular study of a larger lepidopteran group, only data for butterflies might be richer at this point. Over a thousand taxa were sequenced, representing all but one world subfamily (Orthostixinae) and 93 tribes. The geographic coverage, while still rather incomplete for parts of Africa, Southeast Asia, and Australia, is outstanding. The eleven genes represent an enormous character set, of proven utility for reconstructing lepidopteran phylogenies. The phylogenetic analysis is state of the art, especially for such a large number of OTUs. The figures are handsome and a great asset to the manuscript.

One aspect of the study that gives the work extra merit is that many of the authors are among the most active and widely renown geometrid workers. Six of the authors have previously published on the systematics of the Geometridae.

Other commendable aspects of the work include the image album for (Archiearinae, Desmobathrinae and Oenochrominae), the numerous supplemental documents, and the comprehensive list of taxonomic changes organized into a single table. The ICZN rules (and recommendations) and other nomenclatural details appear to have been adhered to.

The authors did a nice job of “folding in” morphological characters (~synapomorphies) into the Discussion. The morphological supplement is a nice addition.

The manuscript is ultradense in taxonomy and as a consequence will not attract readership outside of a small sphere of lepidopterists. But broad appeal has little to do with the importance or value of science—this manuscript will have a very long “shelf life” and can be used as a road map to guide future fieldwork, phylogenetic studies, geometrid classification, and revisionary work.

My only criticism is that the manuscript is still in need of editing and or minor re-tweaking of sentences that are awkward. Someone with strong grammatical skills and or keen eyes could help the effort. A couple different sets of eyes wouldn’t hurt. My Ph.D. student (Kevin Keegan) and I highlighted or annotated issues in the Abstract, Introduction, Methods, Results, and Conclusion, and then only sparsely elsewhere. Annotations have been made in two pdfs, which I will return. Simple yellow-highlighting connotes a place where the team may want to revise. I did not attempt to edit the entire manuscript…that would and could be done much more efficiently with a Word file. (Totally pain with pdfs.)

Be forewarned that the Abstract embedded in the manuscript (beginning on line 33) is in better grammatical shape than the one that appears first in the pdf– that one should be jettisoned.

Minor matters:
Line 35: use familial and tribal as your adjectives, throughout text (do global searches): family/familial and tribe/tribal are mixed in present text

I prefer use of the Oxford comma in scientific writing, especially when the elements of a series are themselves compound. Not sure of the PeerJ policy. Oxford commas absent here.

Line 94-104: I am not a fan of claim staking which is largely what is being done here. Besides these claims and mention of manuscripts in prep reappear in the Discussion and some mentions again in the Conclusions. Geesh.

Line 177 needs redrafting. Not sure what is being said. Wasn’t C01 used in some identifications? If so, say so.

1008. Concluding sentence: “For this taxon and many tribes – old and new – we encourage morphological studies that attempt to find more apomorphies and that include a broader range of taxa.”
Not clear what this taxon refers to as several genera were mentioned in previous sentence. Also while the sentiment in the last sentence is good, the sentence is awkward or uninspiring.

I could suggest as a seed for something a little different.
“We hope the phylogenetic hypotheses shared here will open new paths of inquiry across the Geometridae. Morphological synapomorphies have not yet been identified for many of the re- and newly defined higher taxa circumscribed by our 11-gene data set. Likewise, there is great need, across the family, to begin the work of mapping behavioral and life history attributes to the clades identified in this work.”

·

Basic reporting

The manuscript is written in clear, professional English, but would benefit from careful proof-reading by a native speaker (I've included numerous edits in the attached PDF). Literature is well cited and the context of this paper and taxonomic history of the groups are clearly spelled out. The format of the article conforms to the general structure of a scientific paper, figures, tables and supplementary files are well done and the sequence alignment ("raw data") is shared. A table with GenBank numbers is included, although it still has to be populated for the newly generated sequences. Overall, the amount of detail provided is actually exemplary! The entire publication is self-contained, with the hypotheses postulated in the introduction addressed and answered in the discussion and conclusion; more detailed analysis and taxonomic work has been shifted into separate / pending publications, which is to the benefit of this paper.

Experimental design

The manuscript is well within the aims and scope of this journal, addressing the higher evolutionary relationships between genera, tribes and subfamilies of the mega-diverse moth family Geometridae through molecular phylogenetics. It also implements taxonomic changes based on the results of this study, which is laudable. The need for this kind of study is discussed in the introduction and self-evident from the strong results. The taxon sampling is much broader than typical in the field and was carried out with preference for type species, which is an important aspect often ignored by other studies. Overall, this study is a milestone that greatly advances our understanding and the taxonomy of geometrid moths.

Taxonomic sampling, analyses of molecular data and taxonomic acts are all carried out to a high technical standard. At the same time, some additional data analyses would have been desirable as described below.

The methods and material / sampling is described with great detail and sufficient for replication of the experiment.

Validity of the findings

This study is the by far largest phylogenetic analysis of geometrid moth relationships and greatly advances our understanding of evolutionary relationships due to the unusually large taxon sampling. The methods and data seem robust, with data analyses being statistically sound and well-founded. All conclusions and taxonomic changes are well explained and always linked to and supported by the results. With the manuscript being of a taxonomic nature, it doesn't include speculative elements.

Additional comments

I was very pleased to see that taxonomic and molecular phylogenetic experts teamed up to deliver a very well-rounded research that will be a milestone for the systematics of Geometridae for many years to come. I am particularly impressed by the taxon sampling, the amount of detail provided and the great care taken regarding taxonomic questions. Very well done!

As always, there are some points that could be improved on. While I spell them out, this doesn't mean that the manuscript couldn't be published with no more than some editing by a native speaker - it is good and important as is, with conclusions substantiated by the results! My three points of concern are:

A) Apomorphies in support of the newly defined higher taxa. In all but one case you resort to listing molecular sequence data as diagnostic characters, which might be technically valid, but leaves a taxonomist longing for more. A diagnose should help others to identify unknown taxa as to belong to a taxon like a genus / tribe / subfamily or not. The molecular sequence data listed don't really achieve this, because the data from the other samples are needed, too, and the phylogenetic analyses are as much part of it as the actual sequence data - it's the result of the data analysis that "diagnoses" the species as belonging to a taxon. Furthermore, a diagnosis based on molecular data alone doesn't enable the inclusion / exclusion of other taxa based on morphology, which is the main source of information we have or can readily obtain for many of the taxa not included in your study. For many taxa no molecular-grade specimens are available, which is presumably the reason why you weren't able to include them in the first place. While I'm lamenting the lack of diagnostic morphological characters, I know from own experience how difficult it can be to find such characters and that this might have exceeded the scope of the project. At the same time, looking at the expertise shared between the authors, I wonder who if not you would be in a better position to do so? Hopefully you'll be able to follow up with this one day!

B) The data analyses were carried out carefully and according to current best practice. At the same time, I was surprised to see that only nucleotide data were analysed, mitochondrial and nuclear data combined. Especially for older relationships, it would have been highly advisable to also analyse the data with synonymous changes excluded (i.e., as Degen1 encoded nuclear or amino acid data) and with mitochondrial data excluded. Both synonymous changes and mitochondrial data (20% of this data set) are known to be particularly plagued by compositional heterogeneity due to their rapid evolution, which can mislead phylogenetic analyses. The conflicting result for the placement of Larentiinae (i.e., not as sister of Sterrhinae, but as sister to all remaining Geometridae) seems likely to be a classic case for this problem - more details on this in the comment in the attached text. It's easy to test for (e.g., Euclidean compositional distance tree; SymTest by Lars Jermiin) and easy to counter (e.g., amino acid analysis, recoding with Degen1). Maybe something worth still carrying out to avoid postulating a strongly supported yet possibly / probably incorrect hypothesis - happy to help if need be.

C) Major questions addressed in this paper concern taxa with an exclusive or largely Australian distribution, and places like the Australian National Insect Collection (ANIC) are mentioned in the supplementary text. Yet, as curator of the ANIC Lepidoptera collection, I'm not aware of any of the authors having made an attempt to check our collection (we hold just about every named and unnamed Australian species) or include Australian geometrid experts like Catherine Byrne or Peter McQuillan. While obviously not a must and maybe not always possible, this could have contributed valuable additional information and samples.

In any case, I'm looking forward to seeing the manuscript published - it's a great and important paper as is!

---

## Round 0.2 · Minor Revisions

Dear Dr. Murillo Ramos and colleagues:

Thanks for submitting your manuscript to PeerJ. I have now received one independent review of your work, and as you will see, it is mostly favorable. Nonetheless, the reviewer raised some relatively minor concerns about the manuscript. I agree with the reviewer, and thus feel that his concerns should be adequately addressed before moving forward.

Aside from the criticisms raised by the reviewer’s report, be sure to thoroughly evaluate the marked-up manuscripts kindly provided by the reviewer.

Please ensure that the many grammatical errors are addressed. I advise that you enlist some assistance from an English expert before resubmitting.

Therefore, I am recommending that you revise your manuscript accordingly, taking into account all of the issues raised by the reviewer. I do believe that your manuscript will be ready for publication once these issues are addressed.

Good luck with your revision,

-joe

·

Basic reporting

As stated in my initial review “the manuscript is still in need of editing and or minor re-tweaking of sentences that are awkward. Someone with strong grammatical skills and or keen eyes could help the effort.” Even though authors state that “the text has been changed accordingly” there is much that could be done. I made about 70 suggestions over the first 16 pages and stopped, Mistakes with punctuation, between versus among, that vs which, subject verb agreement. Some of the new text introduced new problems. Elements of a series that contain commas should be separated with semi-colons. Etc.

Experimental design

Excellent.

Validity of the findings

See below.

Additional comments

The data set is enormous and represents a Herculean effort by an expert team of lepidopterists. Despite the analytical limitations and challenges mentioned by reviewer #2 and the writing mechanics, I don’t see a compelling reason not to move forward with acceptance and publication.

Given reviewer Andreas Zwick’s issues, it might be prudent to add one more sentence to the “Searching strategies and model selection” section of the Results that states some of the general uncertainties/misgivings/limitations about the results, and that this study is a best collective guess with the data at hand/methods. In essence, something that serves as a disclaimer given the reservations of Zwick. And then maybe in the Discussion/Conclusion, near the end, mention (again) what needs to be done in terms of taxon sampling, number of genetic markers, and ?analytical methodology to get a stable, robust higher classification for the Geometridae. This would be another way of conveying that this present effort is simply a best effort with the data at hand.

As stated in my initial review “the manuscript is still in need of editing and or minor re-tweaking of sentences that are awkward. Someone with strong grammatical skills and or keen eyes could help the effort.” Even though authors state that “the text has been changed accordingly” there is much that could be done. I made about 70 suggestions over the first 16 pages and stopped, Mistakes with punctuation, between versus among, that vs which, subject verb agreement. Some of the new text introduced new problems. Elements of a series that contain commas should be separated with semi-colons. Etc.

My misgivings are minor. Given the effort and new data being made available, I am in favor of having the publication go forward with but very minor additions, or even none at

---

## Round 0.3 · accepted · Accept

Dear Dr. Murillo-Ramos and colleagues:

Thanks for revising your manuscript to PeerJ, and for addressing the concerns raised by the reviewers. I now believe that your manuscript is suitable for publication. Congratulations! I look forward to seeing this work in print, and I anticipate it being an important resource for research communities studying geometrid systematics and evolution.

Thanks again for choosing PeerJ to publish such important work.

-joe